# Black indium oxide a photothermal $CO_2$ hydrogenation catalyst

Lu Wang[1,2,10 ✉], Yuchan Dong[2,10], Tingjiang Yan[3], Zhixin Hu [4 ✉], Feysal M. Ali[2], Débora Motta Meira[5,6], Paul N. Duchesne[2], Joel Yi Yang Loh[7], Chenyue Qiu[8], Emily E. Storey[7], Yangfan Xu[2], Wei Sun[9], Mireille Ghoussoub[2], Nazir P. Kherani[7,8], Amr S. Helmy[7] & Geoffrey A. Ozin [2 ✉]

Nanostructured forms of stoichiometric $In_2O_3$ are proving to be efficacious catalysts for the gas-phase hydrogenation of $CO_2$. These conversions can be facilitated using either heat or light; however, until now, the limited optical absorption intensity evidenced by the pale-yellow color of $In_2O_3$ has prevented the use of both together. To take advantage of the heat and light content of solar energy, it would be advantageous to make indium oxide black. Herein, we present a synthetic route to tune the color of $In_2O_3$ to pitch black by controlling its degree of non-stoichiometry. Black indium oxide comprises amorphous non-stoichiometric domains of $In_2O_{3-x}$ on a core of crystalline stoichiometric $In_2O_3$, and has 100% selectivity towards the hydrogenation of $CO_2$ to CO with a turnover frequency of $2.44\,s^{-1}$.

[1] School of Science and Engineering, The Chinese University of Hong Kong, Shenzhen, 518172 Shenzhen, Guangdong, China. [2] Solar Fuels Group, Department of Chemistry, University of Toronto, 80 St. George Street, Toronto, ON M5S 3H6, Canada. [3] College of Chemistry and Chemical Engineering, Qufu Normal University, 273165 Qufu, Shandong, China. [4] Center for Joint Quantum Studies and Department of Physics, Institute of Science, Tianjin University, Tianjin, China. [5] CLS@APS, Advanced Photon Source, Argonne National Laboratory, Lemont, IL 60439, USA. [6] Canadian Light Source Inc., 44 Innovation Boulevard, Saskatoon, SK S7N 2V3, Canada. [7] Department of Electrical and Computer Engineering, University of Toronto, Toronto, Canada. [8] Department of Materials Science and Engineering, University of Toronto, 184 College Street, Toronto, ON M5S 3E4, Canada. [9] State Key Laboratory of Silicon Materials and School of Materials Science and Engineering, Zhejiang University, 310027 Hangzhou, Zhejiang, China. [10] These authors contributed equally: Lu Wang, Yuchan Dong ✉email: lwangresearch@gmail.com; zhixin.hu@tju.edu.cn; g.ozin@utoronto.ca

Exploitation of the photothermal effect, specifically to enhance the production of synthetic fuels via gas-phase heterogeneous $CO_2$ hydrogenation, relies upon the nonradiative decay processes of photogenerated electron−hole pairs to simultaneously drive thermochemical and photochemical reactions on the surface of nanostructured materials. Unlike traditional photocatalysis, which is limited to the use of higher-energy photons, photothermal catalysis can theoretically utilize the full wavelength range of the solar spectrum to facilitate surface chemical reactions[1–12].

Optimizing the photonic efficiency of these reactions requires control over the thermal relaxation processes that create high local temperatures and drive surface thermochemical reactions, and the electron−hole generation and separation processes that facilitate surface photochemical reactions[7]. While these two processes appear to be antithetical, requiring one to be optimized at the expense of the other, both are expected to benefit from materials that provide high optical absorption strength over the entire solar spectral wavelength range.

Achieving peak photothermal performance would require the use of black, high surface area nanostructures capable of concurrent thermochemical and photochemical hydrogenation of $CO_2$ to value-added products. Unfortunately, no such materials currently exist.

Our approach to this problem focuses attention on stoichiometric indium oxide ($In_2O_3$). In its pristine form, $In_2O_3$ is a pale yellow thermal insulator with a wide electronic bandgap, absorbing light mainly in the ultraviolet wavelength range and thereby minimizing its ability to function as a photothermal $CO_2$ hydrogenation catalyst[13–18].

Pale yellow $In_2O_3$ can, however, be turned pitch black via thermal hydrogenation at 400 °C. This reduction reaction converts stoichiometric $In_2O_3$ into an oxygen-deficient, non-stoichiometric form, $In_2O_{3−x}$. This black $In_2O_{3−x}$ is shown to be a $CO_2$ hydrogenation catalyst possessing both photothermal and photochemical activity, thereby dramatically transcending the activity and selectivity performance of pale-yellow $In_2O_3$ and, indeed, any other known form of cubic indium oxide[13–16,19].

In brief, we discover that black indium oxide, a non-stoichiometric/stoichiometric heterostructure, denoted $In_2O_{3−x}/In_2O_3$, can enable the photothermal reverse water gas shift reaction (RWGS) under ambient conditions with 100% selectivity. Compared to pale yellow $In_2O_3$, with its CO production rate of 0.78 μmol $h^{−1}$ $m^{−2}$ (19.64 μmol $g^{−1}$ $h^{−1}$) in light, black $In_2O_{3−x}/In_2O_3$ can drive the reaction at 1874.62 μmol $h^{−1}$ $m^{−2}$ (23,882.75 μmol $g^{−1}$ $h^{−1}$), approximately three orders of magnitude greater than both stoichiometric $In_2O_3$ and about three orders of magnitude larger than the best reported RWGS rate of all known indium-oxide-based photocatalysts (Supplementary Fig. 1). The estimated turnover frequency (TOF) is 2.44 $s^{−1}$, which is higher than most photocatalysts and photothermal catalysts for $CO_2$ hydrogenation (Supplementary Table 1). The fact that this synthesis of black indium oxide is straightforward and amenable to scaling speaks well for its application as an industrial photothermal RWGS catalyst.

## Results

**Black indium oxide—synthesis and characterization.** The as-prepared indium oxide nanocrystals (S1) devoid of hydroxide groups was synthesized via thermal dehydroxylation of $In(OH)_3$ nanocrystals at 700 °C for 5 h in air. As-prepared S1 was subsequently treated with hydrogen at different temperatures (i.e., 200, 300 and 400 °C) for 1 h to form $In_2O_{3−x}/In_2O_3$ with different $x$ values, labeled S2, S3 and S4, respectively.

Powder X-ray diffraction (PXRD) patterns of all samples are shown in Fig. 1a. These patterns reveal the presence of the phase pure cubic bixbyite structure type $In_2O_3$ and are devoid of metallic indium, indicating that the black color created during hydrogenation is intrinsic to the indium oxide itself. The cubic bixbyite $In_2O_3$ is a fluorite-type structure $In_2O_3$ with one fourth of the anions missing indicates a periodic structure that produces 25% structural vacancies[20]. The grain sizes calculated from the width of the strongest PXRD peak at 30.6° were found to be 20.6, 30.9, 33.0 and 36.3 nm for S1 through S4, respectively, which confirmed slight size growth with increasing temperature of the hydrogenation.

High-resolution X-ray photoelectron spectroscopy (XPS) of In 3d and Auger LMM 3d electronic transitions served to establish the oxidation state of indium as In(III) (Supplementary Fig. 2). The O 1s core level XPS spectra for all samples could be fit with just two peaks at ~529.2 and ~531.8 eV. These peaks corresponded to lattice oxygen ($O_I$, $InO_6$) and the unsaturated lattice oxygen that generated by the formation of oxygen vacancy ($O_{II}$, $InO_{6−x}$), with the latter population increasing with the temperature of the hydrogenation process, as consistent with the following reaction: $In_2O_3 + xH_2 \rightarrow In_2O_{3−x} + xH_2O$ (Supplementary Fig. 2d).

Since XPS is a surface-sensitive characterization technique, the value of $x$ can be further calculated based on the concentration of oxygen vacancies. As a result, the chemical formulae for S2−S4 are estimated to be $In_2O_{2.8}$, $In_2O_{2.7}$ and $In_2O_{2.63}$, respectively. The absence of a diagnostic hydroxide O 1s XPS peak supports this reaction pathway, rather than an alternative $H_2$ homolysis involving the injection of protons and charge-balancing electrons into the lattice with the concomitant formation of Brønsted hydroxides, $In_2O_3 + xH_2 \rightarrow H_{2x}In_2O_3$.

The thermogravimetric analysis has been performed to simulate the synthetic process of S1 ($2In(OH)_3 \rightarrow In_2O_3 + 3H_2O$) and indicates a very similar value of the weight change (84.94%) for S1 to the theoretical stoichiometric $In_2O_3$ (83.7%), where the very slight excess can be attributed to absorbed water (Supplementary Fig. 3a). Similar measurements were conducted to simulate the synthetic process of S2−S4 (Supplementary Fig. 3b−d) with the observed weight changes of 99.35%, 99.19%, and 98.63%, and indicates the overall formulas of $In_2O_{2.98}$, $In_2O_{2.97}$ and $In_2O_{2.95}$, respectively. Such minor overall weight change confirmed the formation of [O] and the hydrogenation process is a surface treatment and result in the formation of $In_2O_{3−x}/In_2O_3$ heterostructures.

The Brunauer−Emmett−Teller (BET)-specific surface areas of S1–S4 paralleled the monotonic trend of increasing nanocrystal size, registering at 25.15, 15.30, 14.74 and 12.74 $m^2$ $g^{−1}$, respectively, with a pore diameters ranging from 29 to 64 nm (Supplementary Fig. 4).

High-resolution transmission electron microscopy (HRTEM) were employed to investigate the morphology of all samples. The HRTEM images shown in Supplementary Fig. 5 indicate highly crystalline nanomaterials for S1–S3 with the typical (222) facet lattice spacing of ~0.292 nm. By contrast, the images for S4 show the formation of what appear to be amorphous regions, and which seem to be associated with the increased loss of surface O. Lattice regions observed in these images have an average spacing of ~0.296 nm (Fig. 1c and Supplementary Fig. 6). The STEM image also confirmed the imaged nanocrystal is not overlapping with others (Fig. 1d). The average particle sizes of S1 and S4 are calculated as 28.8 and 44.9 nm, respectively (Supplementary Fig. 6e−g).

The optical reflectance spectra (UV-Vis-NIR) for S1–S4 show a trend of gradually increasing absorption, broadening and red shifting of the ultraviolet absorption edge into the visible region with higher temperatures of the hydrogenation (Supplementary Fig. 7). The bandgap of all samples can be then calculated as 2.66, 2.62, 2.54 and 2.36 eV for S1, S2, S3 and S4, respectively.

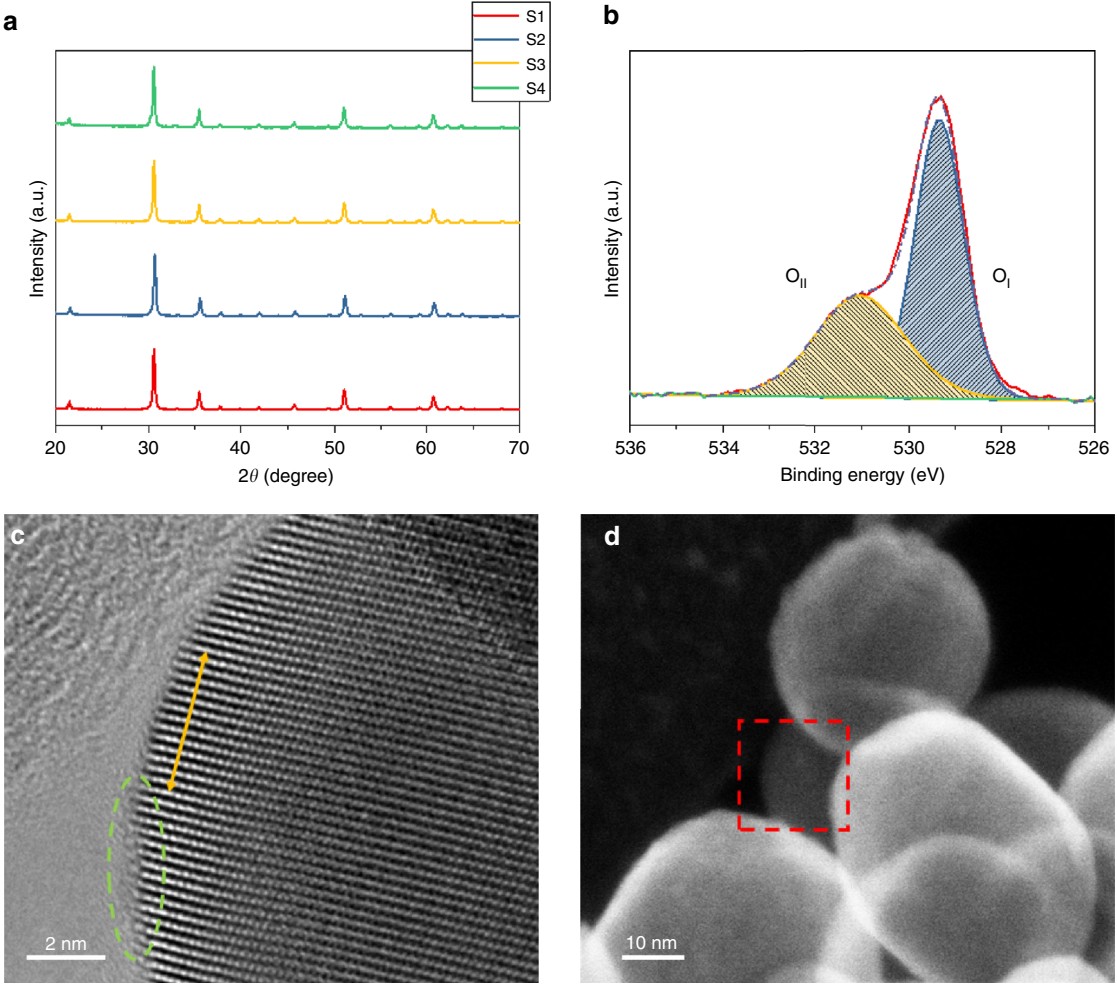

**Fig. 1 Structural and morphological information for $In_2O_{3-x}$/$In_2O_3$ materials S1, S2, S3, S4. a** PXRD patterns of S1–S4. **b** High-resolution O1s core level XPS spectrum of S4. **c**, **d** HRTEM and STEM images of S4 at different magnifications. A dashed green circle indicates an amorphous phase, yellow arrow indicates the measured lattice spacing and the red square indicates the imaged position.

Raman spectra were also recorded for S1–S4. In particular, the mode around 132.3 $cm^{-1}$ is ascribed to the totally symmetrical stretching mode of $InO_6$ octahedral building blocks[21]. With increasing values of $x$ on passing from S1 to S4, this mode undergoes a notable redshift (from 132.3 to 130.0 $cm^{-1}$) and broadening (full-width half maximum increased from 3.24 to 6.21 $cm^{-1}$) (Supplementary Fig. 8)[22,23]. The broadening of peaks also confirmed the amorphization of the surface of black indium oxide. The observed peaks at 132.3, 308.1, 366.5, 497.0 $cm^{-1}$ can be assigned to the phonon vibration modes of the bcc form of $In_2O_3$, while the peak at 366.5 $cm^{-1}$ most likely is associated with the oxygen vacancy which we believe could be considered as the unsaturated lattice oxygen ($InO_{6-x}$)[24]. The ratio between $InO_6$ (132.3 $cm^{-1}$) and $InO_{6-x}$ (366.5 $cm^{-1}$) exhibits a monotonically increasing trend from S1 to S4 and implies an increasing concentration in [O]. Further Raman spectroscopy with different beam intensities were conducted over S3 and S4 to study their photothermal effects. The Raman signal located at ~308 $cm^{-1}$ has been assigned to a vibrational/phonon mode of an $InO_6$ site and is used as the probe/reference signal. The resulting Raman signals for S4 exhibit blue shift with increasing beam intensities (0.00110, 0.00055, 0.00028, and 0.00011 mW $\mu m^{-2}$) gradually shift from 304.18 to 302.90, 302.68 and 300.63 $cm^{-1}$ and indicates an increasing trend of photothermal local temperatures. On the contrary, no significant shift can be observed from S3 and implies minor photothermal effects.

In situ X-ray absorption spectroscopy measurements of S1 under a 3.5% $H_2$/He atmosphere at 400 °C for 2 h indicated partial reduction of In(III) via a shift of the white line peak to lower binding energies (Supplementary Fig. 9). No further shift was observed after an additional 2 h of reductive treatment; however, ex situ measurements of S4 showed more extensive reduction of In(III), resulting in slightly reduced intensity throughout XANES region of the spectrum. There is no sign of metallic indium formation in these spectra, however, which supports the trapping of conduction electrons in mid-gap oxygen vacancy states predominates over the reduction of In(III). Fitting of the Fourier-transformed EXAFS spectra in Supplementary Fig. 10 revealed a slightly lower In–O coordination number of 5.3 relative to the nominal value of 6 (Supplementary Table 2).

In an attempt to delve more deeply into the origin and cause of the aforementioned amorphization processes, an in situ high-resolution environmental transmission electron microscopy (HRETEM) study was performed on stoichiometric $In_2O_3$ nanocrystals under conditions simulating the aforementioned hydrogenation process (Fig. 2). A reference image of a stoichiometric $In_2O_3$ nanocrystal was obtained after annealing and stabilization at 400 °C under an $N_2$ atmosphere for 20 min (Fig. 2a). When exposed to an $H_2$ atmosphere for increasing time periods, the birth and growth of amorphous regions in the nanocrystal can be observed (Fig. 2b–d). The $H_2$-induced removal of O atoms from stoichiometric $In_2O_3$ resulted in

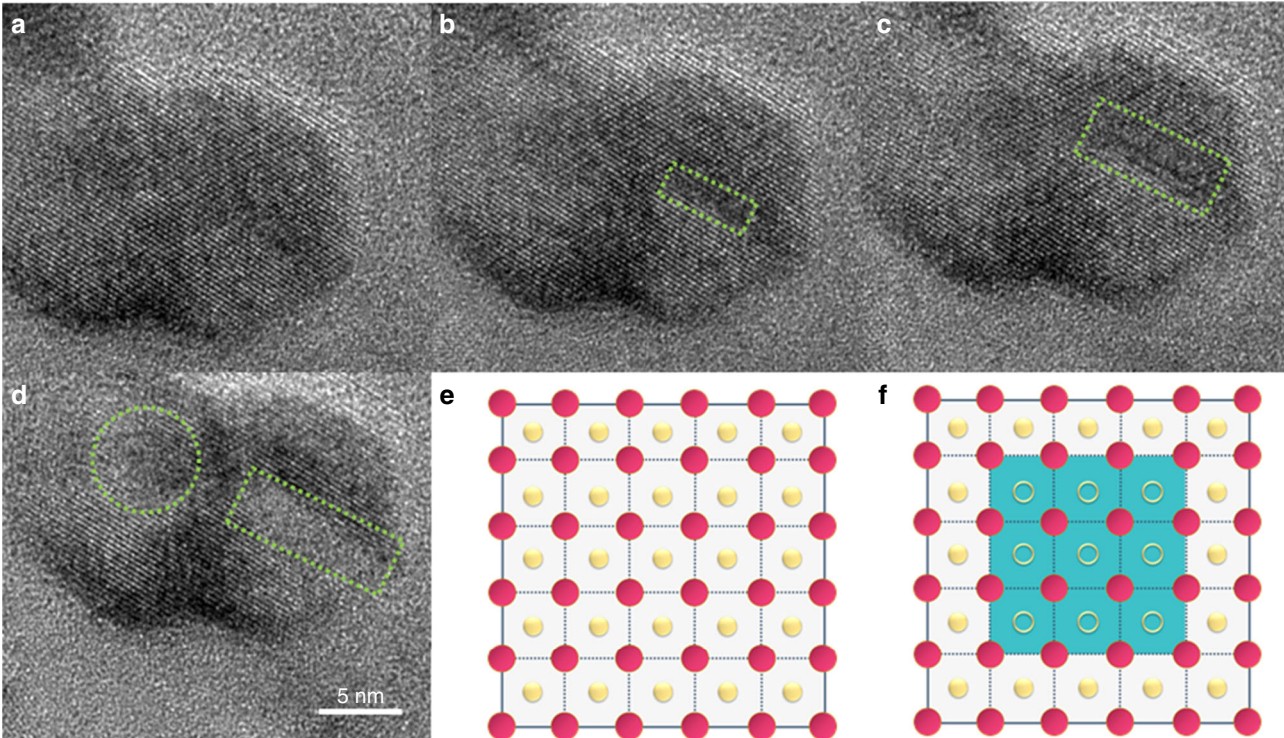

**Fig. 2 In situ high-resolution environmental transmission electron microscopy (HRETEM) observations of the $In_2O_3 + xH_2 \rightarrow In_2O_{3-x} + xH_2O$ process.**
High-resolution images of a stoichiometric $In_2O_3$ nanocrystal (S1) at 400 °C **a** under an $N_2$ atmosphere and then switched to an $H_2$ atmosphere for the following times: **b** 5 min, **c** 10 min and **d** 20 min. Scale bars are the same for all images, green squares indicate the formation of an amorphous phase. Graphical representation of the **e** original and **f** treated $In_2O_3$, wherein blue region, pink dots, yellow dots and yellow circles represent amorphous phase, In atoms, O atoms and [O], respectively.

nonstoichiometric $In_2O_{3-x}$ domains, and thus causing the formation of amorphous phase as illustrated in Fig. 2e, f. The density functional theory calculations performed to analyze the effect of this amorphization process suggests an expansion of the adjacent lattice, proof of which will require atomic resolution TEM (Supplementary Fig. 11 and Supplementary Table 3). The simulated electronic band structures for the simulated $In_2O_3$ are shown in Supplementary Figs. 12 and 13.

**Black indium oxide—photocatalyst evaluation**. It has thus far been established that the black color of hydrogenated $In_2O_3$ stems from the nucleation and growth of amorphous domains of nonstoichiometric $In_2O_{3-x}$ in crystalline stoichiometric $In_2O_3$. Hereafter, black indium oxide will be denoted $In_2O_{3-x}/In_2O_3$. The strong, broad optical absorption of $In_2O_{3-x}/In_2O_3$ across the entire wavelength range of the solar spectrum increases the likelihood that black indium oxide will function as an effective catalyst for the photothermal hydrogenation of $CO_2$. To this end, the photocatalytic activity of black indium oxide on a borosilicate film support was evaluated in a batch reactor. Isotopically labeled $^{13}CO_2$ was used to confirm hydrogenation products originated from carbon dioxide and not adventitious carbon contamination (Supplementary Fig. 14).

The photocatalytic hydrogenation activity of $CO_2$ by samples S1–S4, normalized to their specific surface area, was observed to monotonically increase: 0.78, 1.77, 2.96, 1874.62 μmol $h^{-1}$ $m^{-2}$. This increase paralleled the degree of nonstoichiometry. Notably, the only product observed was CO (Fig. 3a). The S4 sample exhibited the largest conversion rate, being 2403 times higher than that of S1 with TOF of 2.44 $s^{-1}$. The reason for such an impressive photo-enhancement can be attributed to the much

stronger solar energy harvesting ability and photothermal effects of S4 compared to the other samples (Supplementary Figs. 7a and 8), which results in a larger photothermal effect and correspondingly higher catalytic performance. Based on the enclosed thermocouple, the temperatures of S1–S3 are lower than 50 °C, and about 160 °C for S4. The local temperatures of all samples can be estimated from the conversion of $CO_2$ to CO (yield, ppm), where S4 is found to be 262 °C in contrast to S1–S3 which are found to have much lower local temperatures (Supplementary Fig. 15). These results illustrate photocatalysis and photothermal catalysis can be achieved with light irradiation and in this case serve to shift the reaction equilibrium equivalent to one corresponding to 262 °C.

To further evaluate the performance of black indium oxide under more practical conditions, S1 and S4 were tested at 200 °C in a flow reactor with and without light irradiation. As shown in Fig. 3b, S4 (0.10 and 0.14 μmol $h^{-1}$ $m^{-2}$) was much more active towards CO production from $CO_2$ than S1–S3 (ranging from 0.01 to 0.03 and 0.015 to 0.039 μmol $h^{-1}$ $m^{-2}$) both thermocatalytically and photocatalytically (Supplementary Fig. 16a). Increasing temperature causes the CO rates for S4 to continually increase. At 300 °C S4 displayed CO production rates of 160.99 and 38.54 μmol $h^{-1}$ $m^{-2}$. There is a notable photo-enhancement in the CO rate of about 417% under light compared to dark reaction conditions, which is about four times higher than that of the best reported cubic $In_2O_3$-based photocatalyst[13]. The significant photo-enhancement could be caused by the photothermal catalysis. Interestingly, a minor amount of methanol can also be detected in between 250 and 275 °C with light; the detected methanol rates are 1.14 and 1.48 μmol $h^{-1}$ $m^{-2}$, respectively (Supplementary Fig. 16b, c). Formation of methanol at such low $H_2$ concentrations implies a strong hydrogenation

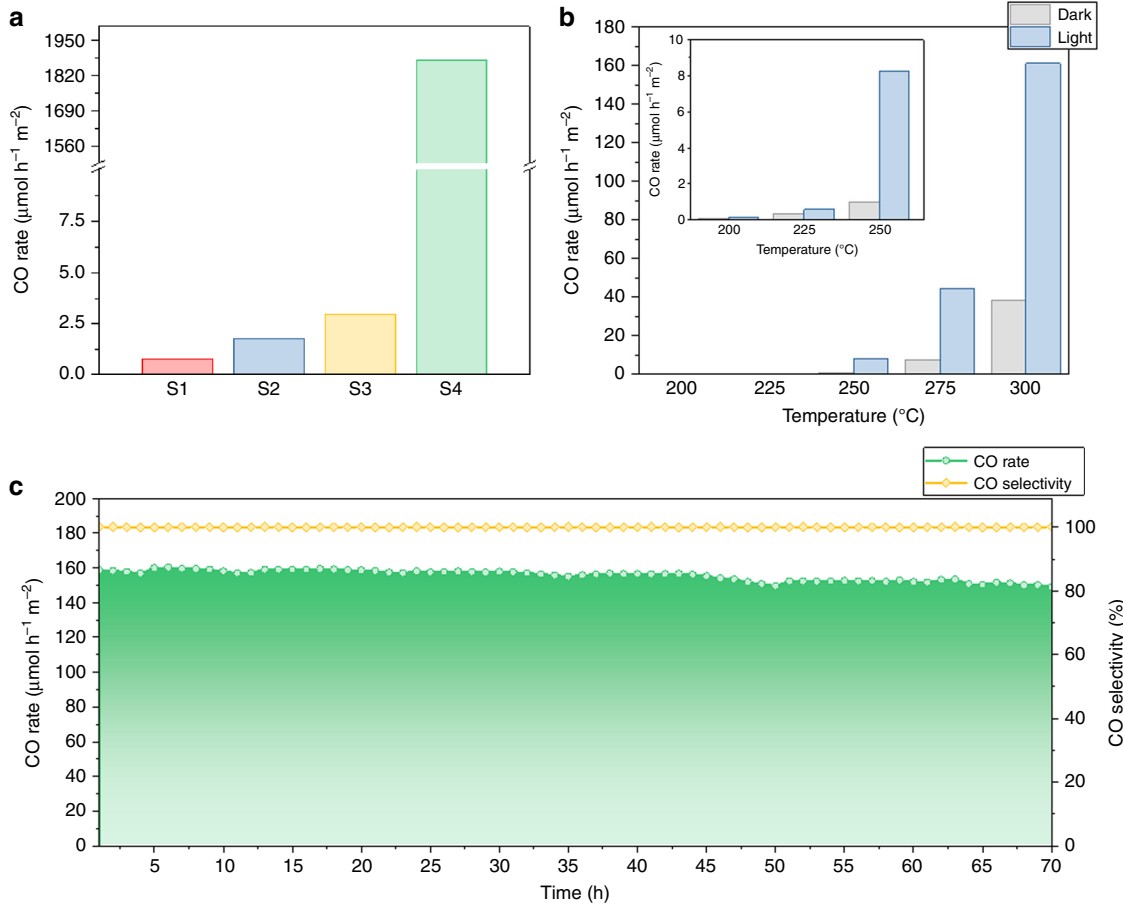

**Fig. 3 Photocatalytic evaluation of black indium oxide. a** Photocatalytic $CO_2$ hydrogenation in a batch reactor. Conditions: $H_2/CO_2$ ratio = 1:1, light intensity = ~20 suns, no external heating and measurement time = 30 min. **b** Catalytic performance for S4 in a flow reactor at different temperatures, both with and without light irradiation; inset is the enlarged view of the catalytic performance for 200, 225 and 250 °C. **c** Stability test for S4 in a flow reactor at 300 °C with light irradiation for 70 h. Conditions for flow measurement: atmospheric pressure, $H_2/CO_2$ ratio = 1:1 with a flow rate of 1 mL min⁻¹ and light intensity of ~8 suns.

ability of black indium oxide. The photoaction behavior of S3 and S4 was examined in a multiwavelength LED photoreactor and exhibited very different catalytic performance (Supplementary Fig. 16d−f). Relative to the activity in the dark, the less photoactive S3 is able to exhibit an increase of 8.8% and 7.7% with irradiation from the UV and blue LEDs, respectively and much less with the green (2.1%) and red (2.2%) LEDs. In stark contrast, S4 exhibits a much stronger photo-enhancement under UV, blue, green or red LEDs (42.6%, 41.0%, 35.9% and 35.2%, respectively). A 70-h photo-stability test for S4 in the flow reactor at 300 °C showed a decrease in activity of only 5% (Fig. 3c). The spent sample of the stability test was probed by PXRD and HRTEM and no obvious change can be identified (Supplementary Fig. 17).

**How does black indium oxide function as a photocatalyst?** We now know that black indium oxide, formed by the thermal hydrogenation of pale yellow indium oxide, is best described as a heterostructure comprising amorphous domains of nonstoichiometric indium oxide $In_2O_{3-x}$ interfaced with crystalline regions of stoichiometric indium oxide $In_2O_3$, which we denote $In_2O_{3-x}/In_2O_3$. We also know that the amorphous phase was caused by the loss of surface oxygen from the $In_2O_3$ phase.

To study the dynamics of photoexcited electrons, in situ photoconductivity measurements under vacuum and experimental conditions for S1 and S4 were performed and shown in Fig. 4a,

b and Supplementary Fig. 18. It is known that, among metal oxides systems, $In_2O_3$ exhibits very long photocurrent relaxation times (minutes to hours), a quality known as persistent conductivity[25,26]. The measurements display photocurrent saturation and decay process lifetimes on the order of minutes. The much faster photo-saturation of the excited electrons for S4 implies a stronger optical absorbance for S4 than S1. While the light was off, the photocurrents of S1 and S4 slowly decayed, with S4 requiring a much longer decay time than S1, which implies a higher population of oxygen vacancy traps for photoelectrons. Furthermore, the longer lifetime of the detected photoelectrons also implies a higher probability for the photoexcited electrons to participate in the reaction and thereby results in a better catalytic performance for S4. The same trend was observed under vacuum conditions, and the resulting $I−V$ plot further confirmed the prolonged lifetime of photoexcited electrons for S4. Moreover, the observed photocurrent is consistent with the observed photocatalytic activity of black indium oxide for $CO_2$ hydrogenation.

In addition, we know that oxygen vacancies [O] and the associated charge-balancing electrons in $In_2O_{3-x}$, arising from the removal of oxygen from $In_2O_3$, cause the color of the material to change from pale yellow to black. The oxygen vacancies [O] and coordinately unsaturated indium In′ sites and oxygen O′ sites in $In_2O_{3-x}$ will exist as mid-gap defect states in the bandgap of $In_2O_3$[27,28]. These states will be respectively situated near the oxide-based valence band and indium-based conduction band, with the charge-balancing electrons occupying mid-gap states, as

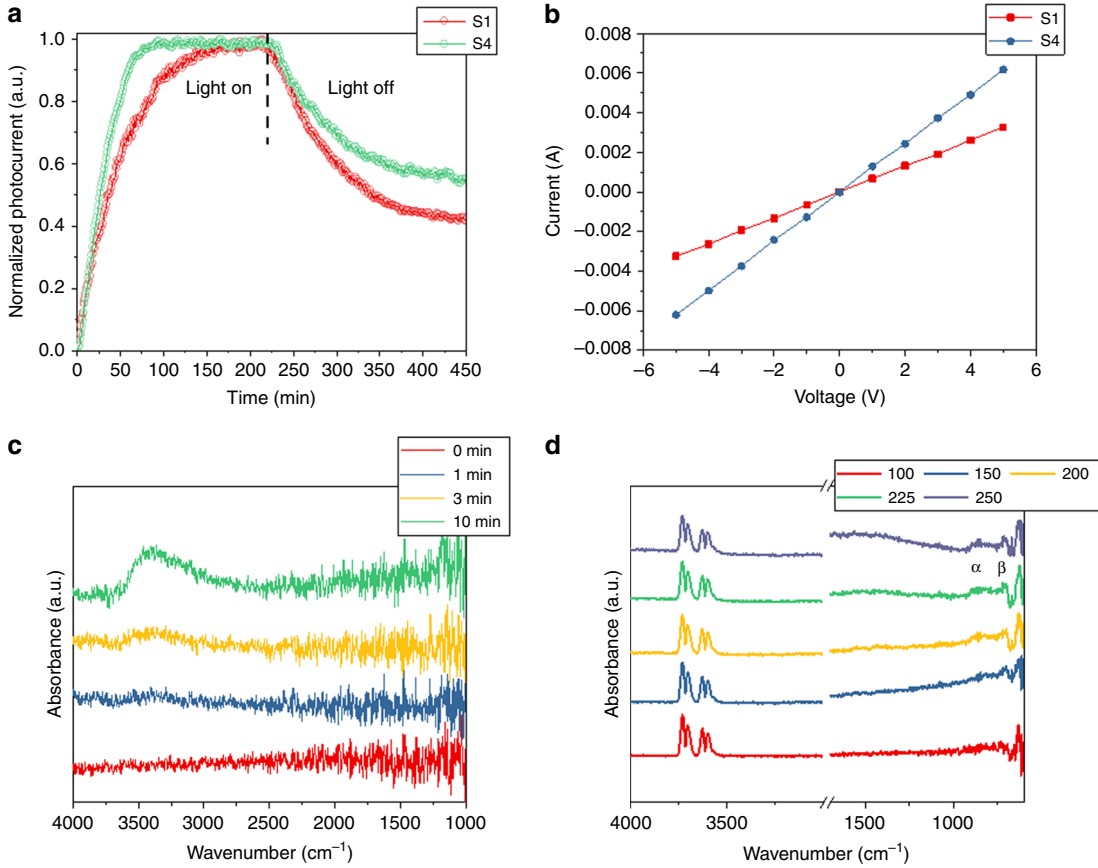

**Fig. 4 Characterization of electronic properties and surface $H_2$ and $H_2$-$CO_2$ chemistry of the samples S1 and S4. a** Photocurrent saturation and decay plot acquired at ~200 °C with a 1:1 ratio of $CO_2$/$H_2$ and under a 100 W LED white lamp. **b** Corresponding in situ $I-V$ plot. In situ DRIFTS spectra of S4 obtained **c** under $H_2$ at room temperature and **d** under both $H_2$ and $CO_2$ (1:1) with increased temperatures. The collected DRIFTS spectra are subtracted by the background signal of S4 obtained under He.

shown in Supplementary Fig. 12e−g. Together, they give rise to optical absorption throughout much of the solar spectral wavelength range.

With this information, we can begin to understand how and why the $In_2O_{3-x}$/$In_2O_3$ heterostructure is well-equipped for photothermal $CO_2$ hydrogenation reactions. Absorption of light across the solar spectral wavelength range causes local heating of $In_2O_{3-x}$/$In_2O_3$ that enables thermochemical conversion of $CO_2$ to CO. Formed via the absorption of light, photogenerated electrons and holes can separate across the interface between $In_2O_3$ and $In_2O_{3-x}$, thereby promoting the photochemical conversion of $CO_2$.

The in situ DRIFTS measurements performed under $H_2$ atmosphere at room temperature revealed homolytic $H_2$ dissociation over black indium oxide (Fig. 4c). The weak peaks around 2900−3700 cm$^{-1}$ and 1100−1200 cm$^{-1}$ can be associated with protonated indium oxide species (In-OH$^+$)[29]. The similarity of these peak positions to those described for In-OH bonds in previous reports, and the disappearance of such peaks on heating above 100 °C, further confirmed the presence of protonated In-O (Supplementary Fig. 19a). The rising baseline in the near infrared spectral range could be caused by free electrons in the conduction band, generated by the homolytic splitting of $H_2$[30]. The absence of indium hydrides in the region of 1100−4000 cm$^{-1}$ implies homolysis of $H_2$, which is different from the former study over $In_2O_{3-x}$(OH)$_y$ that exhibited heterolysis of $H_2$[31].

To further confirm the proposed homolytic $H_2$ splitting mechanism on $In_2O_{3-x}$, solid-state $^1$H MAS NMR spectroscopy was utilized to detect the H-related species in S4 before and after

exposure to $H_2$ at room temperature (Supplementary Fig. 19b). The absence of hydride peak and formation of new peaks at 9.51, 6.91 and 5.66 ppm match well with those of the reference material, $In_2O_{3-x}$(OH)$_y$, leading support to the $H_2$ homolysis pathway (Supplementary Fig. 20a).

While under a $CO_2$ atmosphere at temperatures ranging from 25 to 100 °C, gaseous $CO_2$ fingerprint modes are observed at 3500 −3800 cm$^{-1}$ and 2300−2400 cm$^{-1}$. Two new peaks are also observed at 680 and 825 cm$^{-1}$, and likely signal the activation of absorbed $CO_2$ (Supplementary Fig. 19c, d)[18,32,33] via the insertion of $CO_2$ into O vacancies (Supplementary Fig. 20b).

Insight into the reaction pathway of $CO_2$ hydrogenation has been obtained via in situ DRIFTS under both $H_2$ and $CO_2$ gases (1:1 ratio) (Fig. 4d). Similar to the DRIFTS results obtained under $CO_2$, the results favor a reaction pathway involving the insertion and regeneration of [O], where the $\alpha$ and $\beta$ peaks at 825 and 680 cm$^{-1}$ can be associated with In-O groups originating from the activation and insertion of $CO_2$ (Supplementary Fig. 20c)[34,35].

## Discussion

While $CO_2$ hydrogenation can be driven thermally in the ground electronic state, how can one rationalize the much greater efficiency of the reaction in the excited state?

The relative positions of the valence and conduction bands in the electronic band diagram of $In_2O_3$/$In_2O_{3-x}$, shown in Fig. 5, were obtained via UV-Vis-NIR (Supplementary Fig. 7) and ultraviolet photoelectron spectroscopy (UPS) (Supplementary Fig. 21). One can deduce that following absorption of solar light,

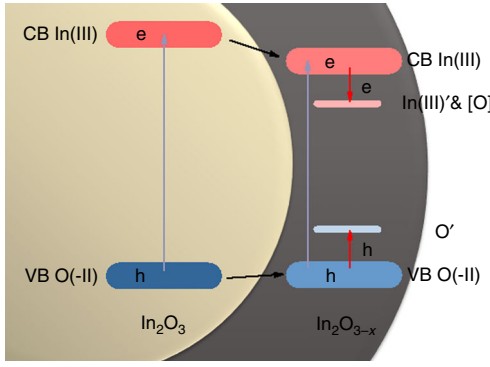

**Fig. 5 Illustration of the electronic band structure.** The $In_2O_{3-x}/In_2O_3$ heterostructure showing the In(III)′, [O] electron-trapping and O′ hole-trapping mid-gap energy states near the CB and VB edges, respectively. Included also in the diagram is the outcome of photoexcitation and relaxation of electrons and holes involving valence, conduction and mid-gap energy states.

photogenerated electrons and holes in the $In_2O_3$ CB can respectively relax to the mid-gap unsaturated In(III)′ and [O] vacancy states near the CB edge and unsaturated O′ near the VB edge in $In_2O_{3-x}$[27]. These states can also be directly populated by band gap excitation and the relaxation of electrons and holes in $In_2O_{3-x}$.

To simulate the formation of surface O vacancies and the reactivity for $CO_2$ hydrogenation, a model $In_2O_3$ (110) surface is used in these calculations (Supplementary Fig. 22). Six possible positions for O vacancies are considered as shown in Supplementary Fig. 23. The calculated energies in Supplementary Table 4 support that the proposed reaction pathways could still occur under solar irradiation conditions. To expand, proton insertion weakens the C−O bond of the adsorbed $CO_2$ at an oxygen vacancy, which enables the formation of CO, thereby reforming $In_2O_3$. Subsequent abstraction of O from $In_2O_3$ by protons to form $H_2O$, reforming the O vacancy and $In_2O_{3-x}$, thereby completes the catalytic cycle. The simulated reaction pathway is represented by the scheme shown in Supplementary Fig. 20d.

Black indium oxide has been found to outperform all known indium-oxide-based photocatalysts based on its activity, selectivity and stability. The calculated turnover frequency of $2.44\,s^{-1}$ is higher than most of the reported photocatalysts and photothermal catalysts in the prior art. It exhibits 100% selectivity towards CO at 300 °C and able to be operated stably for more than 70 h. Its ease of synthesis via hydrogenation of commercially available, pale-yellow indium oxide and shelf-life of more than 9 months (Supplementary Fig. 24) make it highly amenable to scaling for use as an industrial photothermal catalyst.

In this context, the question of cost always arises. For indium, as for all metals, cost depends upon market demand and availability. To amplify on this point, the concentration of terrestrial indium is 0.050 ppm; this concentration is greater than those of silver, which is not considered to be in short supply. With improvements in extraction technology, indium can also be obtained as a by-product from different base metals that include zinc, lead, tin and copper.

The increasing number of geographical locations bearing discovered indium-containing ore deposits, and the positive financial gains of mining companies stemming from greater demand for indium, has encouraged new investments in mining, thereby providing confidence in the stability of indium supply. While still costly at $200−$300 per kg, there currently exist 3000 to 4000 tonnes of stored and commercially available indium, which would

suggest that it can sustain great growth in its use without suffering limitations of supply.

Thus, it will be interesting to see whether black indium oxide proves to be a technologically and commercially viable industrial RWGS photocatalyst.

## Methods

**Chemicals.** Indium(III) chloride (98%) was purchased from Sigma Aldrich and commercial $In_2O_3$ nanocrystal was purchased from Alfa Aesar.

**Synthesis of $In(OH)_3$ nanocrystals.** In a typical synthesis of indium hydroxide, indium(III) chloride (3.6 g, 16.2 mmol, 98%) was dissolved in a 3:1 solution (72 mL) of anhydrous ethanol (Commercial Alcohols) and deionized water. In a separate beaker, a 3:1 mixture of ethanol and ammonium hydroxide was prepared by combining aqueous ammonium hydroxide (18 mL, Caledon, 28–30%) and of anhydrous ethanol (54 mL). The solutions were rapidly combined, resulting in the immediate formation of a white precipitate. The resulting suspension was then immediately immersed in a preheated oil bath at 80 °C and stirred for 30 min. The suspension was then removed from the oil bath and allowed to cool to room temperature. The precipitate was separated via centrifugation and washed five times with deionized water. The precipitate was sonicated between washings to ensure adequate removal of any trapped impurities and then dried overnight at 60 °C in a vacuum oven.

**Synthesis of $In_2O_3$ and hydrogenated samples.** The pale yellow $In_2O_3$ nanocrystal (S1) was obtained via thermal treatment of $In(OH)_3$ nanocrystal in air at 700 °C for 5 h. The hydrogenated samples (S2−S4) were synthesized via treating S1 in tube furnace with $10\%H_2/Ar$ at a flow rate of 120 sccm. The temperatures for treatment were set as 200, 300 and 400 °C for 1 h with an increasing temperature rate of 5 °C per min. When the hydrogenation process was finished, the sample was naturally cooled down to room temperature.

## Data availability

All data are available in the main text or the Supplementary Information.

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

## Acknowledgements

We acknowledge Dr. Kulbir Ghurman for proof reading, Athanasios A. Tountas for estimating the photothermal local temperature, Karl Demmans for the measurement of solid-state $^1$H MAS NMR and Peter Brodersen for the measurement of UPS. G.A.O. acknowledges the financial support of the Ontario Ministry of Research and Innovation (MRI), the Ministry of Economic Development, Employment and Infrastructure (MEDI), the Ministry of the Environment and Climate Change's (MOECC) Best in Science (BIS) Award, Ontario Center of Excellence Solutions 2030 Challenge Fund, Ministry of Research Innovation and Science (MRIS) Low Carbon Innovation Fund (LCIF), Imperial Oil, the University of Toronto's Connaught Innovation Fund (CIF), Connaught Global Challenge (CGC) Fund, and the Natural Sciences and Engineering Research Council of Canada (NSERC). This research used resources of the Advanced Photon Source, an Office of Science User Facility operated for the U.S. Department of Energy (DOE) Office of Science by Argonne National Laboratory, and was supported by the U.S. DOE under Contract No. DE-AC02-06CH11357, and the Canadian Light Source and its funding partners; T.Y. is thankful for financial support from the National Natural Science Foundation of China (21872081), and Natural Science Foundation of Shandong Province (ZR2016BM04); Z.H. thanks the National Science Foundation of China for support through grant no. 11804247.

## Author contributions

L.W. and G.A.O. conceived and designed the experiments. L.W. and Y.D. synthesized the materials, performed in situ DRIFTS and catalytic measurements. Z.H. performed theoretical calculation. T.Y. performed BET measurement. L.W., C.Q., F.M.A. and Y.X. performed and analyzed the HRTEM and in situ HRTEM. D.M.M. and P.N.D. performed and analyzed the in situ XAS. L.W., E.E.S. and A.S.H. performed and analyzed the Raman spectroscopy. W.S. performed UV-Vis-NIR measurement. J.Y.Y.L. and N.P.K. performed and analyzed the in situ photocurrent measurement. F.M.A. estimated the local temperature of sample. M.G. performed TGA measurements. Y.D., L.W. and G.A.O. wrote the manuscript. All authors discussed the results and commented on the manuscript.

## Competing interests

The authors declare no competing interests.
