## [Peer Review File · Nature Communications]

Reviewers' comments:

Reviewer #1 (Remarks to the Author):

The technology reported in this manuscript is of high relevance to the field solar fuels. The process itself, that is, CO₂ reduction into CO, represents a promising pathway towards solar fuels, especially if sunlight-powered and at high efficiency. The material design strategy looks appropriate, creating active sites on indium oxide by H₂-induced formation of substoichiometric domains, leading to substantial activity improvements, although more experimental detail and insight about the reaction are desirable.

Despite the apparent feasibility of the photothermal process based on O-deficient In₂O₃, several aspects on characterisation are supported on either dubious assumptions or data carrying large errors. The derived mechanistic proposal is also debatable to some extent. Specific comments:

1. Structural changes upon H₂ treatment (i). The lattice expansion hypothesis is largely based on HRTEM image analyses giving rise to intensity profiles. The sigmoidal peaks used to determine spacing on these profiles are often unsymmetrical and thus not exactly falling in the midpoint between two valleys (Fig. 1d, 2e), and if other peak maxima are taken, completely different results would be obtained. Therefore, the lattice spacings must be determined after averaging a number of independent measurements, and the standard deviation must be given. Trends from EXAFS are also based on data within experimental error.
2. Structural changes upon H₂ treatment (ii). Lattice expansion is only expected for the few layers closer to the amorphous substoichiometric domains (as shown in Fig. 2g), and most likely gradually changing from that of pristine In₂O₃ to that of In₂O_{3-x}. Therefore, it is difficult to imagine sufficiently perfect (periodical) expanded lattices, and extending throughout sufficiently large crystalline domains, that would give rise to new, genuine and distinct XRD signals as the shoulders shown in Fig. 1b (please also revise lines 80-81, data might be scrambled).
3. Some analysed HRTEM images show superimposed crystallites (Fig. 1c and Suppl. Fig 4), thus resulting in additional error in the intensities used for lattice spacing determination, and in any case representing defective structures.
4. Particle sizes should be statistically determined.
5. Stoichiometries (line 101) derived from XPS only represent near-surface domains. Bulk stoichiometries must be determined from elemental analyses (e.g. ICP).
6. A major O 1s XPS signal is assigned to O vacancies, but this has no logic. If there is no oxygen, how can an intense signal arise? Why is a hydroxide signal discarded? Where would an O 1s signal for In₂O_{3-x} be expected?
7. Kubelka-Munk function plots illustrating bandgap determination are required.
8. Raman data must be commented in more detail.
9. Are DRIFTS plots subtraction spectra against a reference pristine material?
10. The energy states discussed in page 9 are not clearly seen in Suppl. Fig. 10.
11. The proposed homolytic and heterolytic H₂ activation routes must be more clearly described, including unambiguous reaction equations, and commented in relation to eq. [2].
12. How active sites are chosen for TOF determination must be clarified. Which are the differences of extrinsic and intrinsic O vacancies? Are surface-averaged activities calculated over BET data?
13. Activity is enhanced by 3 orders of magnitude from S1 to S4 in batch experiments (Fig. 3a), whereas the difference is much smaller for flow reactors (Suppl. Fig. 15). Why is flow operation activity two orders of magnitude lower (Fig. 3b)?
14. Direct temperature measurements at the catalyst surfaces should be done.
15. Quantum efficiencies and photoaction spectra would also help distinguishing between photonic and thermal activation.
16. Was methanol detected in the gas phase? The GC method utilised to quantify it must be described.
17. The procedure for simultaneous irradiation and heating in the flow reactor must be clearly described, including pictures of the set-up.
18. Conclusions about the actual reported data must be amply included in the Conclusions section, not just future prospects for indium resources.

19. Please complement references 1-8 with these reviews: Green Energy Environ. 2017, 2 (3), 204; Top. Catal. 2016, 59 (15-16), 1268; and these other relevant articles: Adv. Mater. 2016, 28, 3703; Appl. Catal. B, 2018, 235, 186.
20. Experimental section: It is not clear whether hydrogenation of samples was performed on the as-prepared $\text{In}(\text{OH})_3$ or on In_2O_3 (after calcination).
21. How is the catalyst introduced in the batch reactor? Pictures would help.
22. How were irradiances measured?
23. Which is the pressure in the flow reactor?
24. Which is the TEM voltage used?
25. Suppl. Fig. 1. Please indicate the sources and specifications for the P1-P5 materials.

Reviewer #2 (Remarks to the Author):

The authors present an interesting study of the synthesis, characterization, and photocatalytic activity of black indium oxide. They show that it has high photothermal catalytic activity and can selectively hydrogenate $\text{C}=\text{O}$ to CO with a fast turnover frequency due to high optical absorption strength. Black indium oxide is a non-stoichiometric, oxygen-deficient, amorphous, nanostructural form of the oxide deposited on the stoichiometric form of the oxide, resulting in an $\text{In}_2\text{O}_{3-x}$ @ In_2O_3 heterostructure. The authors synthesize black indium oxide by thermal hydrogenating pale-yellow indium oxide (In_2O_3) at 400 °C. They believe that the simple synthesis and superior photocatalytic performance will enable black indium oxide to be used as an industrial photothermal catalyst for the reverse water gas shift reaction. I believe this study is exciting and will be of significant interest to others in the community as well as the wider field. I also believe the paper will influence thinking in the field.

While a large part of the study is experimental, my comments pertain to the computational section of the study, i.e., the results in Figure 2 (f-k), Supplementary Figures 10, 11, and 21, and Supplementary Tables 3 and 4. I do not find the qualitative and quantitative results from the computational work to be sufficiently clear and convincing. However, I believe the work can be improved to strengthen the conclusions and will then be ready for publication in Nature Communications. Please find my comments and suggestions below.

1. The PBE exchange-correlation functional is not accurate enough to study the electronic structure (PDOS and band structure) of an oxide surface. At the very least, a Hubbard correction to PBE (PBE+U) should be applied to study the electronic structure of indium oxide. It would be even better if a hybrid functional, such as HSE, is used to calculate PDOS and band structures, and accurately determine band gaps. While the decrease in band gap with an increase in lattice parameter is probably correctly described by PBE, the amount of the change in the conduction-band edge (0.5 eV, Supplementary Figure 10) and the values of the band gaps (Supplementary Figure 11) should be verified using more accurate functionals.
2. When the authors performed geometry optimizations on the cubic cells in Figures 2(h-k), did they allow the lattice parameter and the volume of the cell to relax in response to the vacancies, and calculate the lattice expansion due to the presence of vacancies?
3. In Supplementary Figures 10-11, the authors indirectly study the effects of oxygen vacancies on the electronic structure of the oxide by incrementally stretching the lattice parameter of stoichiometric In_2O_3 up to 3% and calculating its PDOS and band structure. The underlying hypothesis is that oxygen vacancies stretch the surrounding lattice of In_2O_3 . Why don't the authors calculate the PDOS and band structure of the cubic cells in Figures 2 (h-k) instead to directly study the effects of oxygen vacancies on the

electronic structure of the oxide?

4. The authors have not clarified why they chose the (110) surface of In_2O_3 as the model for calculations of reaction energetics.

5. The authors have calculated reaction energies ranging from 0.69 to 1.26 eV for the conversion of CO_2 to CO (Equation 1) at six different oxygen-vacancy sites on $\text{In}_2\text{O}_3(110)$. Reaction energies are insufficient to determine whether the reaction can occur under solar irradiation. Activation energies should be calculated for the reaction (Equation 1) at the vacancy sites. A similar argument holds for Equation 2, assuming that it is an activated process. The activation energy for thermal desorption of surface oxygen to form gaseous O_2 must be compared to the activation energy for oxygen abstraction by H_2 .

Decision on manuscript NCOMMS-19-33458-T.

Reviewers' comments:

Reviewer #1 (Remarks to the Author):

The technology reported in this manuscript is of high relevance to the field solar fuels. The process itself, that is, CO₂ reduction into CO, represents a promising pathway towards solar fuels, especially if sunlight-powered and at high efficiency. The material design strategy looks appropriate, creating active sites on indium oxide by H₂-induced formation of substoichiometric domains, leading to substantial activity improvements, although more experimental detail and insight about the reaction are desirable.

Despite the apparent feasibility of the photothermal process based on O-deficient In₂O₃, several aspects on characterisation are supported on either dubious assumptions or data carrying large errors. The derived mechanistic proposal is also debatable to some extent.

Author Reply: We thank for the very positive appraisal of the work reported in our paper and deeply appreciate the chance to respond to the comments voiced below.

Specific comments:

1. Structural changes upon H₂ treatment (i). The lattice expansion hypothesis is largely based on HRTEM image analyses giving rise to intensity profiles. The sigmoidal peaks used to determine spacing on these profiles are often unsymmetrical and thus not exactly falling in the midpoint between two valleys (Fig. 1d, 2e), and if other peak maxima are taken, completely different results would be obtained. Therefore, the lattice spacings must be determined after averaging a number of independent measurements, and the standard deviation must be given. Trends from EXAFS are also based on data within experimental error.

Author Reply: The lattice expansion is mainly based on HRTEM and XRD, and careful measurement were performed. To avoid any mistaken leading results, at least 6 lines were measured and the average lattice spacing were calculated.

To confirm the observed results, *in situ* XAS and *in situ* EHRTEM were performed under similar synthetic condition (H₂ and 300 °C). As a result, all of the observed results, including *ex situ* and *in situ* characterization confirmed the lattice expansion.

Based on the fact that the formation of amorphous region and nearby lattice expansion have been shown clearly in Fig. 2, we believe the black indium oxide has the lattice expansion that induced by the formation of amorphous region. The measured lattice spacing also agrees well with the XRD patterns.

More lattice expanded images were added to Fig. S4d-i and the average lattice spacing is calculated as **0.2963 ± 0.0003 nm**.

Supplementary Figure 4. High resolution transmission electron microscopy images of (a) S1, (b) S2, (c) S3 and (d-g) lattice expanded S4, with measured d-spacing of 0.2961 nm, 0.2960 nm, 0.2967 nm and 0.2967 nm, respectively. As a result, the estimated lattice spacing of the expanded (222) is 0.2963 ± 0.0003 nm. STEM images of (h) S1 and (i) S4 with average particle sizes (j) of 28.8 nm and 44.9 nm, respectively.

2. Structural changes upon H₂ treatment (ii). Lattice expansion is only expected for the few layers closer to the amorphous substoichiometric domains (as shown in Fig. 2g), and most likely gradually changing from that of pristine In₂O₃ to that of In₂O_{3-x}. Therefore, it is difficult to imagine sufficiently perfect (periodical) expanded lattices, and extending throughout sufficiently large crystalline domains, that would give rise to new, genuine and distinct XRD signals as the shoulders shown in Fig. 1b (please also revise lines 80-81, data might be scrambled).

Author Reply: We thank for reviewer's careful revision and the PXRD pattern in line 80-81 has been corrected as follow:

As shown by the enlarged PXRD patterns of S4 in Fig. 1b, the (222) and (400) reflections at 30.59° and 35.47° are shifted to 30.21° and 35.09° , respectively, corresponding to a small, yet significant, increase from 0.292 and 0.253 nm to 0.296 and 0.255 nm, Table 1.

We agree that the newly formed PXRD signal is distinct. The size of the prepared black indium oxide nanocrystals is around 30-40 nm. The hydrogenation process is occurring on the surface of In₂O₃ where the lattice expansion occurs. Although the relatively small size of the nanocrystals may result in an enlarged signal of the expanded facets, the major peaks are still coming from the original cubic In₂O₃.

3. Some analysed HRTEM images show superimposed crystallites (Fig. 1c and Suppl. Fig 4), thus resulting in additional error in the intensities used for lattice spacing determination, and in any case representing defective structures.

Author Reply: We thank reviewer's suggestion and remove the description of Fig. S4c for its defective states. The Fig 1c now has been replaced by Fig 1d-f, include 2 HRTEM images of the S4 nanocrystal at different magnifications and an extra STEM image that confirmed no overlapping with other nanocrystals.

The original sentences were corrected as “High-resolution transmission electron microscopy (HRTEM) and scanning transmission electron microscopy (STEM) were also employed to investigate the morphology and lattice structure of all samples. The HRTEM images shown in Supplementary Fig. 4 indicate highly crystalline nanomaterials for S1 through S3. By contrast, the images for S4 show the formation of what appear to be amorphous regions, and which seem to be associated with the increased loss of surface O. Expanded lattice regions were observed in these images, with an average spacing of 0.2963 ± 0.0003 nm that agrees well with the expansion revealed by the aforementioned shoulder on the (222) PXRD reflection, Fig 1c-e and Supplementary Fig. 4d-i. The STEM image confirmed the imaged nanocrystal is not overlapping with others, Fig 1f. The average particle sizes of S1 and S4 are calculated as 28.8 nm and 44.9 nm, respectively, Supplementary Fig. 4h-j.”

Fig. 1 | Structural and morphological information for $\text{In}_2\text{O}_{3-x}@\text{In}_2\text{O}_3$ materials S1, S2, S3, S4. a. PXRD patterns of S1 to S4. b. PXRD patterns for S1 and S4, with evidence of lattice expansion in S4 indicated by the asterisk. c. Normalized pixel intensities of the expanded (222) facets with an average lattice spacing of 0.296 nm. d-e. HRTEM images of S4 at different magnifications. A dashed green circle indicates an amorphous phase and yellow arrow indicates the measured expanded (222) facets. f. STEM of the measured S4 nanocrystals. The red square indicates the imaged position.

4. Particle sizes should be statistically determined.

Author reply: The average particle sizes of S1 and S4 are calculated as 28.8 nm and 44.9 nm, respectively and shown in Fig S4h-j (shown in Q1).

5. Stoichiometries (line 101) derived from XPS only represent near-surface domains. Bulk stoichiometries must be determined from elemental analyses (e.g. ICP).

Author Reply: We appreciate reviewer's suggestion. However, our sample is solely composed of indium oxide, there are only oxygen and indium, which will not give any reasonable results from ICP. On the other hand, our treatment is a surface treatment and XPS is a surface sensitive technique which is suitable for this particular case.

6. A major O 1s XPS signal is assigned to O vacancies, but this has no logic. If there is no oxygen, how can an intense signal arise? Why is a hydroxide signal discarded? Where would an O 1s signal for In_2O_3-x be expected?

Author Reply: We thank the reviewer for raising this important question. The analysis of O1s XPS peaks is a recognized technique for probing oxygen vacancies in materials like In_2O_3 . One of the best ways to view the situation is in terms of a coordination chemistry model where one compares the charge density or effective nuclear charge on ligand O^{2-} bonded to a coordinately saturated 6-coordinate InO_6 versus a coordinated unsaturated InO_x where x is lower than 6, the latter depicting the case of the O vacancy. For the coordinately unsaturated InO_x the O^{2-} ligands will transfer (donate) more of their lone pair electron density to the In^{3+} (acceptor) because of its higher effective nuclear charge relative to the In^{3+} in the coordinately saturated InO_6 . The result is a lower charge density, less screening of the nucleus and a higher effective nuclear charge for the O^{2-} on InO_x resulting in a high energy XPS O1s shift in IP relative to the O^{2-} on InO_6 , which is manifest as a high-energy shoulder on the latter. The proton on the hydroxide causes an even higher energy shift of the XPS O1s IP relative to InO_6 and InO_x . Hence, the diagnostic O vacancy O1s IP occurs at an energy straddled by the dominant lattice O^{2-} IP on the low-energy side and the OH^- IP on the high-energy side. Since XPS is a premier and widely used surface characterization technique, we believe the detected oxygen vacancy sites are mostly located on the surface of the material. The disappearance of OH group is mainly caused by the high temperature calcination at 700 °C for 5 hours.

For the convenience of the referee, related papers on XPS detection of oxygen vacancies in metal oxides are also listed below:

- [1] Lei, F., Sun, Y., Liu, K., Gao, S., Liang, L., Pan, B., Xie, Y. (2014). Oxygen Vacancies Confined in Ultrathin Indium Oxide Porous Sheets for Promoted Visible-Light Water Splitting. *J. Am. Chem. Soc.* *136*, 6826.
- [2] Kim, W., Tak, Y. J., Ahn, B. D., Jung, T. S., Chung, K., Kim, H. J. (2016). High-pressure Gas Activation for Amorphous Indium-Gallium-Zinc-Oxide Thin-Film Transistors at 100°C. *Scientific Reports*, *6*, 23039.
- [3] Jung, Y., Yang, W., Koo, C.Y., Song, K., Moon, J. (2012). High performance and high stability low temperature aqueous solution-derived Li-Zr co-doped ZnO thin film transistors. *J. Mater. Chem.*, *22*, 5390.
- [4] Lee, S., Jeong, Y. E., Lee, D., Bae, J. S., Lee, W. J., Park, K. H., Bu, S. D., Park, S. (2014). Oxygen partial pressure dependent electrical conductivity type conversion of phosphorus-doped ZnO thin films. *J. Phys. D.* *47*, 065306.

7. Kubelka-Munk function plots illustrating bandgap determination are required.

Author Reply: The Tauc plot of Kubelka-Munk function is added into Supplementary Fig 5 and shown as below.

Supplementary Figure 5. Optical reflectance spectra (UV-VIS-NIR) and Tauc plot of Kubelka-Munk function for S1-S4 samples.

8. Raman data must be commented in more detail.

Author Reply: We appreciate the reviewer's suggestion and the Raman section has been edited as follow:

"Raman spectra were also recorded for S1 through S4. In particular, the mode around 132.3 cm^{-1} is ascribed to the totally symmetrical stretching mode of InO_6 octahedral building blocks.²¹ With increasing values of x on passing from S1 to S4, this mode undergoes a notable redshift (from 132.3 to 130.0 cm^{-1}) and broadening (full-width half maximum increased from 3.24 cm^{-1} to 6.21 cm^{-1}), Supplementary Fig. 6.²² The broadening of peaks also confirmed the amorphization of the surface of black indium oxide. The observed peaks at $132.3, 308.1, 366.5, 497.0 \text{ cm}^{-1}$ can be assigned to the phonon vibration modes of the bcc form of In_2O_3 .²⁴"

9. Are DRIFTS plots subtraction spectra against a reference pristine material?

Author Reply: Yes, the DRIFTS are subtracted with the spectra of the black indium oxide in He condition to make sure any newly form peaks are contributed from the reaction intermediates.

10. The energy states discussed in page 9 are not clearly seen in Suppl. Fig. 10.

Author Reply: Supplementary Fig. 10 has been enlarged. Electronic structures for defective cells shown in figure 2 (i-k) are plotted in Supplementary Fig. 11e-g. Several defect states can be found within the gap.

11. The proposed homolytic and heterolytic H_2 activation routes must be more clearly described, including unambiguous reaction equations, and commented in relation to eq. [2].

Author Reply: The procedure of homolytic H_2 splitting is shown in Supplementary Fig 20a and below, whereas hydrogen atom is homolytic split over In-O sites to form 2 protons (or protonated In-OH^+) and 2 electrons. The absence of In-H^- around 1400 cm^{-1} also confirmed the homolytic H_2 dissociation.

Due to the uncertainty of the products, there are no accurate reaction equations for such splitting process, it could only be described as $\text{In}_2\text{O}_3-x + y\text{H}_2 \rightarrow \text{H}_2\text{In}_2\text{O}_3-x$. The equation 2 is describing the process of [O] regeneration, has no relationship with H_2 dissociation.

The information of heterolytic H_2 dissociation can be found in our previous paper (<https://doi.org/10.1002/anie.201904568>) and the dissociation process is shown below. Clearly, the major difference for these 2 pathways is the presence of In-H^+ , which can help us to confirm the H_2 dissociation pathway.

12. How active sites are chosen for TOF determination must be clarified. Which are the differences of extrinsic and intrinsic O vacancies? Are surface-averaged activities calculated over BET data?

Author Reply: As shown in Supplementary Information, Page2, line 48. We use 3 different models to calculate the TOF, including utilize all [O] as active sites, utilize external (newly generated from the treatment) [O] as active sites and utilize [O] in amorphous region as active sites.

Based on the lattice structure of cubic In_2O_3 , it has 25% intrinsic [O] in its lattice structure, the extrinsic [O] can be calculated as overall concentration of [O] – 25%.

BET data is used to determine the number of active sites.

13. Activity is enhanced by 3 orders of magnitude from S1 to S4 in batch experiments (Fig. 3a), whereas the difference is much smaller for flow reactors (Suppl. Fig. 15). Why is flow operation activity two orders of magnitude lower (Fig. 3b)?

Author Reply: There is no external thermal supply for batch experiment, which means the temperatures of different catalyst are solely based on their absorption of light source. The black indium oxide has a much stronger UV-Vis-NIR absorption, which implies a stronger photothermal effect (higher temperature under same condition) and result in huge different in photothermal catalytic performance.

On the other hand, there is an external thermal supply for flow reactor, which can make the temperature difference much smaller and result in less enhanced performance.

The flow and batch experiment have very different reaction conditions, including different light intensity, sample loading, irradiated area, and flow rate (batch does not flow). As a result, the very different catalytic performance can be expected.

14. Direct temperature measurements at the catalyst surfaces should be done.

Author Reply: The temperatures shown for flow reactor (200-300 °C, Fig. 3) are obtained from the thermocouple that direct attached the catalyst, which also confirmed the mixed photonic and thermal catalytic performance. We have tried to detect the temperature for batch photothermal measurement via IR camera and thermocouple, unfortunately, the observed temperature seems to be strongly influenced by the light source and too low to be believed (~100 °C). On the other hand, temperature calculated from the thermodynamic equilibrium is much more accurate that can reflect the exact temperature for active site of the catalyst and we believe this is the better way to report the real temperature rather than a meaningless low temperature.

15. Quantum efficiencies and photoaction spectra would also help distinguishing between photonic and thermal activation.

Author Reply: We appreciate the reviewer's suggestion.

The QE of batch reaction (20 suns, 30min, ~1mg catalyst loading) for S4 with the rate of $1874.62 \mu\text{mol h}^{-1}\text{m}^{-2}$ ($23,882.75 \mu\text{mol g}^{-1}\text{h}^{-1}$) is estimated as 0.04% based on the following equation and Xe lamp spectrum.

$$\text{EQ} = \text{mol of produced CO} / \text{mol of photon} = 1.19414\text{E-}05 / 0.029171077 = 0.04\%$$

The photoaction plot has been performed and added to the manuscript as “The photoaction behavior of S4 was examined in a LED photoreactor and exhibited relatively similar CO enhancement under UV, blue, green or red LED beam source (42%-35%), Supplementary Fig. 15d.”

Supplementary Figure 15. (a) Catalytic performance of S1 and S4 in a flow reactor at 200 °C. (b) GC spectra at different temperature with light and (c) Photocatalytic performance at 250 °C and 275 °C for both CO and methanol. (8.27 and 44.56 $\mu\text{mol h}^{-1}\text{m}^{-2}$ for CO; 1.14 and 1.48 $\mu\text{mol h}^{-1}\text{m}^{-2}$ for methanol) (d) Photoaction of S4 in LED reactor. Test conditions: 300 °C, light intensity of ~ 5 suns, pressure of ~ 40 psi, gas ratio of CO_2 and H_2 is 1:1 at 2 ml/min.

16. Was methanol detected in the gas phase? The GC method utilised to quantify it must be described.

Author Reply: All products of catalytic CO_2 reduction, including methanol, were detected in the gas phase. Product gas concentrations were calculated by GC software (PeakSimple) using integrated peak areas and the corresponding calibration curves. Production rates measured in the flow reactor were calculated by mass balance, taking into account the current temperature, pressure, and volumetric flow rate, then normalized to the mass of catalyst used in the reactor. The separation of gaseous products was performed using a 2 m HayeSep D capillary column. In each run, the column was first preheated to a temperature of 40 °C and held for 2 min, then heated from 40 °C to 80 °C and held at 80 °C for 2 min, after that the column is heat to 100 °C and held at 100 °C for 10 min before it was heated to 220 °C and hold for another 5 min (total run time: 37 min + 13 min oven cool down, heating rate 10 °C/min).

17. The procedure for simultaneous irradiation and heating in the flow reactor must be clearly described, including pictures of the set-up.

Author Reply: Detail procedure has been shown in SI, page 5, line 178-189. The light performance and dark performance are tested separately to make sure there is no influence by the light irradiation. A picture of flow reactor has been shown as below.

18. Conclusions about the actual reported data must be amply included in the Conclusions section, not just future prospects for indium resources.

Author Reply: We appreciate reviewer's suggestion and the first paragraph of conclusion has been edited as follow:

Black indium oxide has been found to outperform all known indium-oxide-based photocatalysts based on its activity, selectivity and stability. The calculated turnover frequency of 2.12 s^{-1} is higher than most of the reported photocatalysts and photothermal catalysts in the prior art. It exhibits 100% selectivity towards CO at 300 °C and able to be operated stably for more than 70 hours. Its ease of synthesis via hydrogenation of commercially available, pale-yellow indium oxide and shelf-life of more than 6 months, **Supplementary Fig. 22**, make it highly amenable to scaling for use as an industrial photothermal catalyst.

19. Please complement references 1-8 with these reviews: Green Energy Environ. 2017, 2 (3), 204; Top. Catal. 2016, 59 (15-16), 1268; and these other relevant articles: Adv. Mater. 2016, 28, 3703; Appl. Catal. B, 2018, 235, 186.

Author Reply: We have included the aforementioned papers.

20. Experimental section: It is not clear whether hydrogenation of samples was performed on the as-prepared $\text{In}(\text{OH})_3$ or on In_2O_3 (after calcination).

Author Reply: In_2O_3 is the one used for hydrogenation.

The original sentence was corrected as "The hydrogenated samples (S2-S4) were synthesized via treating S1 in tube furnace with 10% H_2/Ar at a flow rate of 120 sccm."

21. How is the catalyst introduced in the batch reactor? Pictures would help.

Author Reply: Catalyst is drop cast onto the borosilicate film (Fig. S23) which will be input into the batch reactor during the reaction. Picture is shown below.

22. How were irradiances measured?

Author Reply: The light intensity was measured with Spectra-Physics Power meter (model 407A). The measured reading is about 2 W with a 1cm² mask and giving a light intensity of 2 W/cm². It is known that 1 sun = 0.1 W/cm², thus the light intensity is about 20 suns. Same method is applied to flow reactor and the light intensity is calculated as 8 suns.

23. Which is the pressure in the flow reactor?

Author Reply: Atmospheric pressure.

24. Which is the TEM voltage used?

Author Reply: 300 kV.

25. Suppl. Fig. 1. Please indicate the sources and specifications for the P1-P5 materials.

Author Reply:

References are added to Fig S1's caption.

P1 is commercial In₂O₃ that been used in this manuscript,

P2 is In₂O₃ prepared from thermal decomposition of In(NO)₃ at 300 °C,

P3 is In₂O₃-xOHy nanocrystal superstructured nanorod used the method from reference 1,

P4 is rhombohedral In₂O₃-xOHy nanocrystal used the method from reference 2,

P5 is In₂O₃-xOHy nanocrystal used the method from reference 1.

The related references are listed below:

1. Joule, 2018, 2, 1369-1381. (cubic In₂O₃-xOHy nanocrystal and superstructure)
2. Nature Communication, 2019, 10, 2521. (rhombohedral In₂O₃-xOHy nanocrystal)

Reviewer #2 (Remarks to the Author):

The authors present an interesting study of the synthesis, characterization, and photocatalytic activity of black indium oxide. They show that it has high photothermal catalytic activity and can selectively hydrogenate C_0 to CO with a fast turnover frequency due to high optical absorption strength. Black indium oxide is a non-stoichiometric, oxygen-deficient, amorphous, nanostructural form of the oxide deposited on the stoichiometric form of the oxide, resulting in an $In_2O_{3-x}@In_2O_3$ heterostructure. The authors synthesize black indium oxide by thermal hydrogenating pale-yellow indium oxide (In_2O_3) at 400 °C. They believe that the simple synthesis and superior photocatalytic performance will enable black indium oxide to be used as an industrial photothermal catalyst for the reverse water gas shift reaction. I believe this study is exciting and will be of significant interest to others in the community as well as the wider field. I also believe the paper will influence thinking in the field.

While a large part of the study is experimental, my comments pertain to the computational section of the study, i.e., the results in Figure 2 (f-k), Supplementary Figures 10, 11, and 21, and Supplementary Tables 3 and 4. I do not find the qualitative and quantitative results from the computational work to be sufficiently clear and convincing. However, I believe the work can be improved to strengthen the conclusions and will then be ready for publication in Nature Communications. Please find my comments and suggestions below.

Author Reply: We thank for the very positive appraisal of the work reported in our paper and deeply appreciate the chance to respond to the comments voiced below.

1. The PBE exchange-correlation functional is not accurate enough to study the electronic structure (PDOS and band structure) of an oxide surface. At the very least, a Hubbard correction to PBE (PBE+U) should be applied to study the electronic structure of indium oxide. It would be even better if a hybrid functional, such as HSE, is used to calculate PDOS and band structures, and accurately determine band gaps. While the decrease in band gap with an increase in lattice parameter is probably correctly described by PBE, the amount of the change in the conduction-band edge (0.5 eV, Supplementary Figure 10) and the values of the band gaps (Supplementary Figure 11) should be verified using more accurate functionals.

Author Reply: We agree that the PBE functional would underestimate the bandgap of the In_2O_3 structure. The HSE method is computationally expensive for our cell. The PBE+U method was applied to get a better reproduction of the electronic structure, with corrections of $U_{In_s}=11eV$ and $U_{O_p}=8eV$. The direct gap of bulk In_2O_3 is calculated to be 2.31 eV. After stretching lattice parameters by 1% ~ 3%, the bandgap gradually decreases to 1.82eV, qualitatively consistent with the previous PBE calculation. Optimized lattice parameters by PBE+U method are 0.5 % shorter than PBE method. No obvious structural distortion was found. The Figure S11 is now updated with new results.

Supplementary Figure 11. Simulated band structure for In_2O_3 crystals. (a-d) Valence band maxima are set as the zero level. The width of the direct gap at the gamma point is marked in each panel. The bandgap decreases from 2.31 eV to 1.82 eV when the lattice is stretched by 3%. (e-g) Calculated band structures for defective structures shown in figure 2(i-k). Defect states in red are found in the middle of the bandgaps, which may enhance the adsorption of light.

2. When the authors performed geometry optimizations on the cubic cells in Figures 2(h-k), did they allow the lattice parameter and the volume of the cell to relax in response to the vacancies, and calculate the lattice expansion due to the presence of vacancies?

Author Reply: The volume of the cubic cell is not an essential value since it is not very sensitive to the existence of a single point defect. The O vacancy only causes distortion of surrounding region, as illustrated in figure 2 (g-k). The size of the cell is optimized in the calculation. It is found that changes of lattice parameters were less than 0.04% with 1~3 adjacent O atoms removed. The almost unaltered volume proves that the selected cubic cell is large enough to prevent the interaction between defects from another periodic cell.

3. In Supplementary Figures 10-11, the authors indirectly study the effects of oxygen vacancies on the electronic structure of the oxide by incrementally stretching the lattice parameter of stoichiometric In_2O_3 up to 3% and calculating its PDOS and band structure. The underlying hypothesis is that oxygen

vacancies stretch the surrounding lattice of In_2O_3 . Why don't the authors calculate the PDOS and band structure of the cubic cells in Figures 2 (h-k) instead to directly study the effects of oxygen vacancies on the electronic structure of the oxide?

Author Reply: PDOSs or band structures of figure 2(h-k) structures are not shown in the previous manuscript basing on following reasons: 1) The PBE method underestimates the band gap and PBE+U introduces artificial constant U. For both cases the position of the defect state may be incorrect. 2) Those structures are just specific cases for the amorphous region. The real structure can be much more complicated, then the electronic structural would also be different.

We add band structures for cells in figure 2(h-k) to the Fig S11e-g. It shows that some defect states appear in the middle of the gap. It doesn't mean that any amorphous structure has the same feature. (Figure S11 is attached in Q1)

4. The authors have not clarified why they chose the (110) surface of In_2O_3 as the model for calculations of reaction energetics.

Author reply: The lattice spacing of (222) facets are used to estimate the lattice expansion. It is also an important clue for the surface plane. In-O lines from the (110) plane form patterns in HRETEM images. We add a new Supplementary figure 21 to show those In-O lines.

Supplementary Figure 21. Top view of the In_2O_3 (110) surface. Red and brown spheres represent O and In atoms respectively. (222) facets are marked by blue dashed lines. The average distance between adjacent (222) facets is calculated to be 0.297nm for a defect-free surface, consistent with the measured lattice spacing. This lattice spacing confirms that the surface of the sample is In_2O_3 (110).

5. The authors have calculated reaction energies ranging from 0.69 to 1.26 eV for the conversion of CO_2 to CO (Equation 1) at six different oxygen-vacancy sites on In_2O_3 (110). Reaction energies are insufficient to determine whether the reaction can occur under solar irradiation. Activation energies

should be calculated for the reaction (Equation 1) at the vacancy sites. A similar argument holds for Equation 2, assuming that it is an activated process. The activation energy for thermal desorption of surface oxygen to form gaseous O₂ must be compared to the activation energy for oxygen abstraction by H₂.

Author reply: The conversion reaction of CO₂ to CO can be achieved with the oxygen-vacancy and adsorbed H atoms. The whole reaction can be divided into 4 steps. Step 1, the CO₂ fills into an oxygen-vacancy on the surface. This is an exothermic process with energy decrease of 0.84eV. Step 2, an incoming H atom helps to break one C-O bond, forming CO* and OH* on the surface. The activation energy is calculated to be 1.31eV. Step 3, The CO molecule detaches from the surface, this reaction is endothermic with an activation energy of 0.38eV. Step 4, the OH* and another H atom forms a H₂O molecule and detach from the surface, creating a new oxygen-vacancy. This reaction has an activation energy of 1.87eV. Activation energies are acquired with the nudged elastic band method. The reaction pathway is updated in the manuscript.

Supplementary Figure 19. Proposed reaction pathways for (a) H₂ dissociation, (b) CO₂ adsorption, (c) CO₂ hydrogenation and (d) stimulated CO₂ hydrogenation pathway. Energy barriers for H insertion, CO desorption and H₂O desorption are 1.31eV, 0.38 eV and 1.87 eV respectively. The CO₂ adsorption process is exothermic with energy decrease of 0.84eV.

Reviewers' comments:

Reviewer #1 (Remarks to the Author):

Significant additional work has been included in this resubmission. Some interesting photo(thermo)catalytic data actually reinforce the design strategy for black indium oxides. However, the characterisation studies regarding O defects are still highly debatable. Specific comments:

1. The lattice expansion, as determined from HRTEM image analyses, still relies on intensity profiles carrying significant error. The main argument in this regard is based on the HRETEM measurements shown in Fig. 2e, which do not exactly fall in the midpoint between two valleys. If other peak maxima are taken, for example the previous one (taking the 6th maximum instead of the 7th), completely different results would be obtained. The HRETEM images are (understandably) of lower quality than those obtained from regular HRTEM. But then, expansion in the S4 sample is also claimed on the grounds of regular HRTEM on a four different measurements, giving an average of 0.2963 nm. I suspect that the measured domain corresponds to a standard In₂O₃ showing (2 2 2) spacing (some works also report 0.296 nm or similar spacings, see Appl. Organometal. Chem. 2007, 21, 909; New J. Chem. 2019, 43, 10689; or J. Catal. 2017, 355, 26). The result of a number of analogous measurements for the other samples (S1, S2 and S3) should be done under identical conditions, and the obtained data compared.

2. Trends from EXAFS have to be taken with care since they are based on data increments within experimental error.

3. Lattice expansion is only expected for the few layers closer to the amorphous substoichiometric domains (as shown in Fig. 2g), and most likely gradually changing from that of pristine In₂O₃ to that of In₂O_{3-x}. Therefore, it is difficult to imagine sufficiently perfect (periodical) expanded lattices, and extending throughout sufficiently large crystalline domains, that would give rise to new, genuine and distinct XRD signals as the shoulders shown in Fig. 1b.

4. Regarding stoichiometries, it is true that XPS is a surface-sensitive technique, and thus, it only represent near-surface (nanometer range depths) domains, but XRD is expected to provide information on deeper domains (micrometer range), much closer to the bulk scale, and the authors support some of their conclusions on XRD data. Ideally, EDX elemental analyses from SEM observations could be performed to determine the composition up to micrometer depths, and then, compared to bulk stoichiometries determined from true bulk ICP elemental analyses.

5. The O 1s XPS signals assigned to O vacancies is also present for S1, and hence, the pristine indium oxide material, as synthesised by thermal dehydroxylation under air at 700 °C, also contains O vacancies, specifically 28.91% of all O, according to Supplementary Fig. 2. The authors claim that this equates to 25% intrinsic O vacancies in S1. What does this mean? Which is then the actual surface stoichiometry for S1? Moreover, the fit of deconvoluted components in O 1s XPS must be superimposed to the experimental trace, to check that no extra signal components are required. The results from calculation method used to determine stoichiometries from O vacancy percentages (Supplementary Information, page 3) must be compared to the direct quantification of relevant In and O XPS signals, giving rise to surface In/O ratios.

6. Regarding the rationale of higher binding energy for O atoms in unsaturated InO(6-y) environments, I do not find it obvious that the central indium atom will have a higher effective nuclear charge. H₂ treatment does reduce the material, and it is expected that it is indium itself which becomes reduced, at least partly and locally in the case of those unsaturated In atoms, which are thus not anymore In³⁺, but In^{(3-z)+}, and also thus maintain electroneutrality. Partly reduced In, as In^{(3-z)+}, will exert a smaller electrostatic attraction towards lone pair O electrons, as compared to In³⁺. This would compensate for the smaller number of coordinating O atoms. Furthermore, the generation of O vacancies is expected to result in a continuum of differently unsaturated octahedral environments, and not to two clearly distinguishable types of O atoms in two totally different structural environments, giving rise to two distinct XPS signals. Although a totally different approach, please check the invariability of O 1s binding energy across different oxidation states of a metal in different oxides in this example: Journal of Electron Spectroscopy and Related Phenomena 2004, 135, 2-3, 167.

7. Fig 4c,d and related text. It should be clearly stated (i) which material is measured (S4?), and (ii) that DRIFTS plots are the result of subtraction spectra against a reference pristine material.
8. Regarding how TOFs were determined, it must be clarified how the number of surface O atoms were calculated, and how (if that was the case) BET surface area data were used to do so.
9. Activity is enhanced by 3 orders of magnitude from S1-3 to S4 in batch experiments (Fig. 3a), although the amount of oxygen vacancies is not so much different. Convincing explanations on the extremely different performances of H₂-treated materials, e.g. S3 and S4, should be provided, ideally by showing full photothermocatalytic data also for S3 in order to compare them to those for S4.
10. More detailed discussion on the experiments under monochromatic LED light should be given. Ideally, analogous data should be obtained for less photo-active materials (e.g. S3), in order to prove that S4 is truly and uniquely activated by light across a wide range of wavelengths. Irradiances of each LED source, and photonic efficiencies would be also informative.
11. Pictures of the irradiation setups and reactors should be included.
12. Light intensity measurement method must be included in the Supplementary information.
13. Pressure in the flow reactor should be specified.

Reviewer #2 (Remarks to the Author):

I have read the point-by-point response letter and the revised manuscript, and feel that the points I raised in my previous review on the computational component of the paper have been satisfactorily addressed by the authors.

Reviewers' comments:

Reviewer #1 (Remarks to the Author):

Significant additional work has been included in this resubmission. Some interesting photo(thermo)catalytic data actually reinforce the design strategy for black indium oxides. However, the characterisation studies regarding O defects are still highly debatable. Specific comments:

Author Reply: We thank the reviewer for the overall positive appraisal of the work reported in our paper and deeply appreciate being given the chance to respond to the comments and questions voiced, dealt with chronologically below, all of which proved to be of great value in producing a much high quality paper, which we hope will be deemed acceptable and enable the paper to be published in Nature Communications.

1. The lattice expansion, as determined from HRTEM image analyses, still relies on intensity profiles carrying significant error. The main argument in this regard is based on the HRETEM measurements shown in Fig. 2e, which do not exactly fall in the midpoint between two valleys. If other peak maxima are taken, for example the previous one (taking the 6th maximum instead of the 7th), completely different results would be obtained. The HRETEM images are (understandably) of lower quality than those obtained from regular HRTEM. But then, expansion in the S4 sample is also claimed on the grounds of regular HRTEM on a four different measurements, giving an average of 0.2963 nm. I suspect that the measured domain corresponds to a standard In₂O₃ showing (2 2 2) spacing (some works also report 0.296 nm or similar spacings, see Appl. Organometal. Chem. 2007, 21, 909; New J. Chem. 2019, 43, 10689; or J. Catal. 2017, 355, 26). The result of a number of analogous measurements for the other samples (S1, S2 and S3) should be done under identical conditions, and the obtained data compared.

Author Reply: We do understand reviewer's concern, but all our results, including HRTEM, EHRTEM, XRD, and XAS agree with each other, and confirm the lattice expansion of the black indium oxide. Our characterization methods are similar to the following recent lattice strain papers (Science, 2016, 354, 1031; Science, 2019, 363, 870; ACS Nano, 2019, 13, 4761; and J. Am. Chem. Soc, 2013, 135, 14691) that are mainly based on HRTEM and XRD results. We believe all our results are correct and well represent the lattice expansion of the black indium oxide.

We also recognized that the quality of XRD patterns for Appl. Organometal. Chem. 2007 is quite low and we cannot really tell if there is any overlapping peak or not. They did not mention any XRD peak position in the manuscript and commented the observed lattice spacing as "The HRTEM image shown in Fig. 3(b) reveals an inter-planar distance of 0.296 nm **CLOSE** to the (222) lattice spacing of the cubic phase In₂O₃." indicates the authors understand the observed lattice spacing is not equal to the real lattice spacing 0.292 nm. The other 2 papers (New J. Chem. 2019, 43, 10689; or J. Catal. 2017, 355, 26) are all heterostructures with TiO₂ with a very weak (222) peak signal in their XRD pattern. It is not surprising that the heterostructure can induce lattice expansion, and even if there is a diagnostic expansion signal in XRD pattern, it will likely be buried in the noise.

Nevertheless, to respect the reviewer's understandable concern, we have performed another HRTEM measurement study over S1 to S3 and find the standard lattice spacing for (222) is ~0.292 nm for all of them and confirmed the lattice expansion of S4.

Supplementary Figure 5. TEM images of (a1) S1, (b1) S2 and (c1) S3. High resolution TEM images of (a2-a5) S1, (b2-b5) S2 and (c2-c5) S3 with average lattice spacing's of 0.2918 ± 0.0007 nm, 0.2917 ± 0.0008 nm and 0.2915 ± 0.0005 nm, respectively.

2. Trends from EXAFS have to be taken with care since they are based on data increments within experimental error.

Author Reply: We agree with the reviewer's suggestion and remove conclusion on bond length.

3. Lattice expansion is only expected for the few layers closer to the amorphous substoichiometric domains (as shown in Fig. 2g), and most likely gradually changing from that of pristine In_2O_3 to that of $\text{In}_2\text{O}_{3-x}$. Therefore, it is difficult to imagine sufficiently perfect (periodical) expanded lattices, and extending throughout sufficiently large crystalline domains, that would give rise to new, genuine and distinct XRD signals as the shoulders shown in Fig. 1b.

Author Reply: It has been shown in previous publications that surface modification of a material can induce lattice strain (including both expansion and compression), and the minor change in the interface/surface of the material can induce a much stronger and distinct XRD signals than our result (shown below). We think the distinct XRD signal of the lattice expanded black indium oxide should represent the true state of the material.

Figure 4. Powder XRD spectra of the Pd–Rh nanoparticles. Shown in panel a are spectra for Pd–Rh NPs, Pd–Rh nanoboxes, and Rh cubic nanoframeworks with monometallic bulk spectrum peak positions of both metals given for comparison. The 220 peak region is magnified and shown in panel b.

Figure 6. In panel a is a magnified STEM/EDX image of the shell region. In panel b is an HRTEM image of the shell of a single Pd–Rh nanobox. Panel c gives a FFT contrast-enhanced HRTEM image of the shell showing curvature of {200} planes with grid overlaid of the set of *d*-spacing measurements, which are plotted in panel d. Averaged *d*-spacing along and across the shell are given in panels e and f, respectively.

Source: *J. Am. Chem. Soc.*, 2013, 135, 14691 (the huge XRD signal is actually caused by the outer shell expansion)

4. Regarding stoichiometries, it is true that XPS is a surface-sensitive technique, and thus, it only represent near-surface (nanometer range depths) domains, but XRD is expected to provide information on deeper domains (micrometer range), much closer to the bulk scale, and the authors support some of their conclusions on XRD data. Ideally, EDX elemental analyses from SEM observations could be performed to determine the composition up to micrometer depths, and then, compared to bulk stoichiometries determined from true bulk ICP elemental analyses.

Author Reply: We appreciate reviewer’s suggestion. However, to the best of our knowledge, ICP (mainly ICP-MS) is a bulk characterization technique, which would not be suitable to quantify the black indium oxide that formed via the surface hydrogenation. In addition, ICP cannot detect oxygen from a metal oxide. In brief, to prepare an ICP sample, metal oxide needs to be completely dissolved into the acid solution and the resulting suspension is used to measure the ICP. In this case, ICP is not a useful technique for our work. Secondly, EDX has been widely used for elemental mapping/distribution, not for quantitative analysis of single component metal oxides. Quantitative EDX is an unreliable technique, due to the sample preparation (water suspension may influence the surface oxygen composition), pretreatment (ozone treatment to remove surface carbon contaminates) and uncertainty of the penetration depth of the beam, the resulting information can easily cause misunderstanding.

On the other hand, TGA is another well accepted bulk quantitative analytic technique for stoichiometric studies. The TGA measurement that simulates/mirrors the synthetic process of S1 from $\text{In}(\text{OH})_3$ is performed. Despite the fast dehydration process at the beginning, the relative slow weight dropping process indicates the generation of oxygen vacancy and the latter weight gain process implies the oxidation of the generated oxygen vacancy. The final stabilized stage has an overall weight of 84.9% of $\text{In}(\text{OH})_3$, which is really close to the theoretical value of 83.7%. ($2\text{In}(\text{OH})_3 \rightarrow \text{In}_2\text{O}_3 + 3\text{H}_2\text{O}$) The resulting difference in weight could be caused by the surface absorbed water. (The simulated synthetic process of S1 is added to the manuscript)

Similar TGA measurements were performed to simulate the synthetic process of S2 to S4 and provide a final weight of 99.35%, 99.19%, and 98.63%, which confirmed the removal of surface oxygen and generation of [O]. Furthermore, the overall formula of S2 to S4 can be estimated as $\text{In}_2\text{O}_{2.98}$, $\text{In}_2\text{O}_{2.97}$ and $\text{In}_2\text{O}_{2.95}$, respectively. However, as the TGA is measuring the overall change in the weight, our hydrogenation treatment is actually a surface treatment, it will be inappropriate to use the results that obtained from TGA to estimate the actual chemical formula for S2 to S4. As a result, we did not include these TGA information to avoid misunderstanding.

5. The O 1s XPS signals assigned to O vacancies is also present for S1, and hence, the pristine indium oxide material, as synthesised by thermal dehydroxylation under air at 700 oC, also contains O vacancies, specifically 28.91% of all O, according to Supplementary Fig. 2. The authors claim that this equates to 25% intrinsic O vacancies in S1. What does this mean? Which is then the actual surface stoichiometry for S1? Moreover, the fit of deconvoluted components in O 1s XPS must be superimposed to the experimental trace, to check that no extra signal components are required. The results from calculation method used

to determine stoichiometries from O vacancy percentages (Supplementary Information, page 3) must be compared to the direct quantification of relevant In and O XPS signals, giving rise to surface In/O ratios.

Author Reply:

Based on the concentration of [O] (28.91%), the formula of S1 can be estimated as $\text{In}_2\text{O}_{2.9}$, which is close to In_2O_3 .

The reason we claim S1 is similar to commercial standard In_2O_3 is based on the similarity between S1 and commercial In_2O_3 in XPS and Raman spectroscopy, and TGA measurement.

- For XPS, there is only 0.58 % difference in the concentration of the oxygen-related compounds. As a result, we have assumed that S1 is In_2O_3 . The ratio between In and O for commercial In_2O_3 , S1 to S4 are 1.16, 1.15, 1.14, 1.13, and 1.11, respectively. The result of a wide scan/survey spectra (atomic concentration of different elements, usually has 10% error) from XPS are not as accurate as high resolution XPS spectra. The actual In/O for commercial In_2O_3 (1.16) may vary from 1.42 to 0.94, which is too big to be considered as a meaningful result.

(Left) Survey spectra of S1 and commercial In_2O_3 , (right) high resolution O1s for S1 (upper) and commercial In_2O_3 (lower).

	In-O	[O]
S1	71.09 %	28.91 %
Commercial In_2O_3	71.67 %	28.33 %

Previous publication has shown that the In_2O_3 has a bixbyite structure, a fluorite-type structure with $\frac{1}{4}$ of the anions missing; a periodic structure that produces structural vacancies (inherent/intrinsic oxygen vacancies, 25%). Therefore, in the crystalline structure, all indium cations are surrounded by 6 O atoms and 2 structural vacancies, which we believe will be $\sim 25\%$ [O]. (The DFT simulation presented in our response to Q6 below provides additional evidence and credible support for the existence of structural [O])

Figure 2. Structure of crystalline In_2O_3 (bixbyite).

Source: Chem. Matter. 2014, 26, 5401.

- The Raman spectroscopy of S1 shows very similar result to previous reports and a commercial In_2O_3 . The peak at 132 and 366 cm^{-1} are representing the signal of In-O and [O]. Therefore, the [O]/In-O of S1 to S4 are 0.135, 0.149, 0.161 and 0.192, implying an increasing trend of the concentration of [O] from S1 to S4.

Raman spectra of S1 and commercial In_2O_3 .

Supplementary Figure 8. (a) Raman spectra for S1-S4. (b) Ratio between the peaks at 132 and 366 cm^{-1} . Peak at 132.3 cm^{-1} (S1) red shifts to 130.0 cm^{-1} (S4) with increasing x value. The obtained full-width half-maximum of S1 to S4 are 3.24 , 3.61 , 4.82 and 6.21 cm^{-1} , respectively. The relative intensities for peak at $\sim 132 \text{ cm}^{-1}$ (representing In-O) are 0.790 , 0.785 , 0.622 and 0.274 for S1 to S4; relative intensities for peak at $\sim 366 \text{ cm}^{-1}$ (representing [O]) are 0.107 , 0.117 , 0.100 and 0.053 for S1 to S4. Therefore, the ratios between [O] and In-O for S1 to S4 are 0.135 , 0.149 , 0.161 and 0.192 , which imply an increasing trend of the concentration of [O] from S1 to S4.

- As shown in Q5, the TGA is another well-accepted characterization technique for estimating the oxygen composition for a metal oxide. The final stabilized process for S1 has an overall weight of 84.9% , which is close to the theoretical value of 83.7% . ($2\text{In}(\text{OH})_3 \rightarrow \text{In}_2\text{O}_3 + 3\text{H}_2\text{O}$)

Therefore, Fig S2 is corrected with experimental spectra shown as red dash-dot. The O1s spectra of S1 and commercial In_2O_3 are added as well.

Supplementary Figure 2. (a) Wide scan of S1-S4. High resolution XPS spectra of (b) In 3d, (c) Auger LMM of In and (d) O 1s for all samples. High resolution XPS spectrum of O 1s for (e) S4 and (f) S1 and commercial In_2O_3 , which can be fitted into 2 peaks at ~ 529.2 eV and ~ 531.8 eV, representing lattice oxygen and oxygen vacancy. The concentration of [O] for commercial In_2O_3 , S1 to S4 are 28.33%, 28.91%, 31.74%, 34.97% and 37.34%, respectively. Based on the very similar composition for In-O and [O], the S1 is assumed to be In_2O_3 .

6. Regarding the rationale of higher binding energy for O atoms in unsaturated $\text{InO}(6-y)$ environments, I do not find it obvious that the central indium atom will have a higher effective nuclear charge. H2 treatment does reduce the material, and it is expected that it is indium itself which becomes reduced, at least partly and locally in the case of those unsaturated In atoms, which are thus not anymore In^{3+} , but $\text{In}(3-z)^+$, and also thus maintain electroneutrality. Partly reduced In, as $\text{In}(3-z)^+$, will exert a smaller electrostatic attraction towards lone pair O electrons, as compared to In^{3+} . This would compensate for the smaller number of coordinating O atoms. Furthermore, the generation of O vacancies is expected to result in a continuum of differently unsaturated octahedral environments, and not to two clearly distinguishable types of O atoms in two totally different structural environments, giving rise to two distinct XPS signals. Although a totally different approach, please check the invariability of O 1s binding energy across different oxidation states of a metal in different oxides in this example: Journal of Electron Spectroscopy and Related Phenomena 2004, 135, 2–3, 167.

Author Reply: This is a good point which can be viewed in different ways. In the context of XPS detection and assignments of [O] this all depends on the effect an [O] has on neighboring oxygen atoms in its coordination sphere. These [O] exist as mid-gap electronic states which can be detected by different

forms of spectroscopy as well as the effect they have on catalytic activity especially ones that are light assisted as in the case of our study.

To amplify in the language of a chemist, on passing from the stoichiometric cubic In_2O_3 to the non-stoichiometric cubic $\text{In}_2\text{O}_{3-x}$ there is a decrease in the oxygen coordination number associated with the two crystallographically distinct indium sites from 6 in the former to 5 or lower with x in the latter – as a result of this decrease there is a concomitant and compensating increase in charge transfer from the remaining oxygen's to the indium which can induce a corresponding blue shift in the energy of the XPS O1s ionization potential as observed in practice – the effect of this charge redistribution on reducing the coordination number of InO6 sites presumably must overwhelm any associated decrease in the effective oxidation state of the indium site originating from the non-stoichiometry.

The mid-gap O vacancy states serve other roles related to the photocatalytic behavior of the SFLP active sites - they tune the Lewis acidity, Lewis basicity and geometry, which impacts the chemical reactivity of proximal O-In-O-In-[O] active sites as well as serving to trap and lengthen the lifetime of photo-generated electron-hole pairs, which enhances photo reactivity compared to competitive photo physical relaxation processes as observed in the case of our study.

To further strengthen our point, a DFT simulation has been performed to study the Bader charge and the corresponding energy state associated with the [O]. Specifically, the Bader analysis for surface atoms on (a) In_2O_3 (110) and (b) $\text{In}_2\text{O}_{3-x}$ (110) with a [O] is performed. Red and brown spheres represent O and In atoms, respectively. Values of the charge redistribution are shown versus the neutral state of each element. Blue numbers indicate electron accumulation while red numbers indicate electron loss. A green square marks the surrounding structure pertinent to the [O]. According to the charge redistribution, the formation of [O] causes the nearby In1 to In4 atoms to gain electron density while concomitantly the neighboring O1 loses some.

In addition, DFT simulated O1s energy states of the vacancy-free In_2O_3 and vacancy-containing $\text{In}_2\text{O}_{3-x}$ surface were computed for each O atom (shown below as x-axis in (a)) and labeled as shown in (b). The calculated average O1s state energy is -510.56 eV for the vacancy-free surface and calibrated to the experimental binding energy of In-O at 529.2eV. The formation of [O] causes the O1s state energy, especially for O1 atom, to decrease by 1.22eV. There are 12 surface O atoms in the simulated supercell

and based on symmetry considerations of the O, it is seen that 6 types of [O] can be created on the surface. It is found that for each type of [O], the average O1s state energy is lower than the pristine surface with an energy difference in the range -0.04eV to -0.35eV. Therefore, the simulation concurs well with the experimental XPS results and provides additional support for the proposal that the presence of surface [O] in $\text{In}_2\text{O}_{3-x}$ does indeed result in an increase in the binding energy of O1s electrons, as observed in the XPS.

Furthermore, it is important to note that the O1s XPS spectra have been utilized as evidence of [O] for In_2O_3 -based material for more than 10 years and much great work has been reported based on the presence and effect of [O]. References to ten related In_2O_3 -based O1s XPS papers are shown below. Based on our experimental results, together with the DFT modeling and much documented prior work, we are forced to the inescapable conclusion that [O] vacancies can be detected and quantified by high resolution O1s XPS spectra.

In_2O_3 -based work with [O] vacancies analyzed by O1s XPS:

1. Langmuir 2006, 22, 9380.
2. Nature Material 2011, 10, 45.
3. Cryst. Growth Des. 2012, 12, 4104.
4. Sci. Rep. 2013, 3, 1021.
5. J. Mater. Chem. A, 2014, 2, 5490.
6. J. Am. Chem. Soc. 2014, 136, 6826.
7. Cryst. Res. Technol. 2015, 50, 884.
8. J. Phys. Chem. C 2016, 120, 9874.
9. Applied Catalysis B: Environmental 2017, 218, 488.
10. J. Mater. Chem. C, 2018, 6, 4156.

7. Fig 4c,d and related text. It should be clearly stated (i) which material is measured (S4?), and (ii) that DRIFTS plots are the result of subtraction spectra against a reference pristine material.

Author Reply: We have correct the caption of Fig 4c&d to “*Operando* DRIFTS spectra of S4 obtained c. under H₂ at room temperature and d. under both H₂ and CO₂ (1:1) with increased temperatures. The collected DRIFTS spectra are subtracted by the background signal of S4 that obtained under He.”.

8. Regarding how TOFs were determined, it must be clarified how the number of surface O atoms were calculated, and how (if that was the case) BET surface area data were used to do so.

Author Reply: We agree with the reviewer’s point and add the following information into SI. We also updated the actual exposed facet and the calculated TOF were reassessed and corrected.

Simulated exposed facet with 12 surface oxygen atoms, where red and brown colors denote oxygen and indium atoms.

Area of simulated exposed facet $14.57\text{\AA} \times 10.31\text{\AA} = 1.50 \times 10^{-18}\text{ m}^2$

Number of surface O atoms per square meter = $8 \times 10^{18}\text{ m}^{-2}$

Therefore the overall surface O atoms can be calculated by multiplying the number of O atoms per square meter with the specific surface area obtained from BET measurements.

9. Activity is enhanced by 3 orders of magnitude from S1-3 to S4 in batch experiments (Fig. 3a), although the amount of oxygen vacancies is not so much different. Convincing explanations on the extremely different performances of H₂-treated materials, e.g. S3 and S4, should be provided, ideally by showing full photothermocatalytic data also for S3 in order to compare them to those for S4.

Author Reply: The huge difference in the catalytic performance is mainly caused by the color/optical absorption of the catalyst (Fig. S5 and S23) and is not solely dependent on the concentration of defects.

The enhancement in catalytic performance is most like coming from the photothermal effect, which provides the thermal energy that is powering the reaction. In thermocatalysis, an increment of 10 degrees results in an exponential enhancement in the catalytic performance. This will also apply to

photothermal catalysis and can result in a large difference between a poor and an excellent photothermal catalyst. In this case of our study, under the same reaction conditions, the black catalyst (S4) is expected to exhibit a much stronger photothermal effect and higher local temperature than the white-pale yellow catalysts (S1-S3) and thereby experience a huge performance photo-enhancement as observed in practice.

This explanation has been added to the manuscript “The reason for such an impressive photo-enhancement can be attributed to the much stronger solar energy harvesting ability of S4 compared to the other samples (Supplementary Fig. 7a), which results in a larger photothermal effect and correspondingly higher catalytic performance.”

Additional catalytic performance results were also added into the manuscript (Fig S16a).

Supplementary Figure 16. (a) Catalytic performance of S1 to S4 in a flow reactor at 200 °C. (b) GC spectra of S4 at different temperatures with light and (c) Photocatalytic performance of S4 at 250 °C and 275 °C for both CO and methanol. (8.27 and 44.56 $\mu\text{mol h}^{-1} \text{m}^{-2}$ for CO; 1.14 and 1.48 $\mu\text{mol h}^{-1} \text{m}^{-2}$ for methanol) (d) Catalytic performance of S3 and S4 at 300 °C in LED reactor without any light irradiation. Photoaction behavior of (e) S3 and (f) S4. Test conditions: 300 °C, light intensity of ~ 5 suns, pressure of

~40 psi, gas ratio CO₂:H₂ = 1:1 at 2 ml/min. The peak wavelengths of UV, blue, green and red LED sources are 365, 440, 525 and 625 nm, respectively. Pictures of reactor setup (g) batch reactor, (h) flow reactor and (i) LED reactor.

10. More detailed discussion on the experiments under monochromatic LED light should be given. Ideally, analogous data should be obtained for less photo-active materials (e.g. S3), in order to prove that S4 is truly and uniquely activated by light across a wide range of wavelengths. Irradiances of each LED source, and photonic efficiencies would be also informative.

Author Reply: We agree with the reviewer's point and use S3 as the less photo-active material.

The original sentences were corrected as follow "The photoaction behavior of S3 and S4 was examined in a multi-wavelength LED photoreactor and exhibited very different catalytic performance, **Supplementary Fig. 16d-f**. Relative to the activity in the dark, the less photoactive S3 is able to exhibit an increase of 8.8% and 7.7% with irradiation from the UV and blue LEDs and much less with the green (2.1%) and red (2.2%) LEDs. In stark contrast, S4 exhibits a much stronger photo-enhancement under UV, blue, green or red LED LEDs (42.6%, 41.0%, 35.9% and 35.2%)."

The peak wavelength of UV, blue, green and red LEDs are 365, 440, 525 and 625 nm, respectively. The external quantum yields at these wavelengths can be estimated as 0.044%, 0.045%, 0.030% and 0.026%, respectively.

The improved Fig S16 is shown in Q9.

11. Pictures of the irradiation setups and reactors should be included.

Author Reply: We agree with reviewer's comment and add the images into manuscript. (Images in Q9)

12. Light intensity measurement method must be included in the Supplementary information.

Author Reply: We agree with reviewer's comment and add the method into the manuscript.

"The light intensity was measured with a Spectra-Physics Power meter (model 407A). The measured reading is about 2 W with a 1 cm² mask and giving a light intensity of 2 W/cm². It is known that 1 sun = 0.1 W/cm², thus the light intensity is about 20 suns in the batch reactor. The same method is applied to the flow reactor and the light intensity is calculated as 8 suns."

13. Pressure in the flow reactor should be specified.

Author Reply: We agree with reviewer's comment and add the pressure used in the flow reactor into the manuscript.

"Conditions for flow measurement: atmospheric pressure, H₂/CO₂ ratio = 1:1 with a flow rate of 1 mL min⁻¹ and light intensity of ~8 suns."

Reviewers' comments:

Reviewer #1 (Remarks to the Author):

After carefully going through the authors's responses and modifications on the manuscript, I am pleased to check that comparative experiments under monochromatic light have provided evidence for the largely enhanced activity of the material showing clearly superior light absorption. This stronger light absorption is presumably related to the amount of defects, but this does not seem to be a directly proportional relationship, and extended discussion on structure-activity aspects would be appreciated, since it is a central topic of this work. Explanations on O vacancies are convincing in general terms, and the reason why they actually exist in stoichiometric In₂O₃ is clear. However, assignment of spectroscopic signals directly to O vacancies is inaccurate. In addition, brief structural description would help the reader. Despite the noticeable improvement in the soundness of the manuscript, the lattice expansion issue is still unsupported with current data, it cannot be accepted as it is, and moreover, it is not significantly relevant to the main conclusions. Specific comments:

1. The discussion on lattice expansion. Truly, lattice strain can be used to engineer materials, as in the references suggested by the authors, but always on a short-range. Lattice expansion is only expected for the few layers closer to the amorphous substoichiometric domains, and hence, it is difficult to imagine sufficiently perfect (periodical) expanded lattices, and extending throughout sufficiently large crystalline domains, that would give rise to new, genuine and distinct XRD signals as the shoulders shown in Fig. 1b.

2. Still on lattice expansion, HRTEM measurements carry significant errors. In Fig. 2e, intensity maxima do not exactly fall in the midpoint between two valleys, and if other peak maxima are taken, for example the previous one (taking the 6th maximum instead of the 7th), completely different results would be obtained. Higher (even atomic-scale) resolution would be required for lower error and higher reliability of the data.

3. The lattice spacing data determined for S4 based on the yellow lines in Suppl. Fig. 6 might be biased since such yellow lines do not appear to be orthogonal to the intensity fringes.

4. Lines 177-8: "gradually increasing values 0.292, 0.293, 0.294 and 0.296 nm on passing from S1 through S4" are claimed, but these data are for dynamic measurements on a sample of stoichiometric In₂O₃ under H₂ atmosphere directly on the EHRTEM instrument, not for samples S1 through S4.

5. Regarding (2 2 2) spacing values, it is true that a value of around 0.292 nm is consistent with crystallographic data, but it is still intriguing why the spacing remains essentially invariable, or even shows a decreasing trend, when going from S1 through S2 and finally S3, as shown in Suppl. Fig. 5 (0.2918±0.0007 nm, 0.2917±0.0008 nm and 0.2915±0.0005 nm, respectively).

6. The theoretical adjacent distance between (2 2 2) facets is reported to be 0.297 nm in Suppl. Fig. 22. This does not match experimental values, as in comment 5 above (0.292 nm).

7. Quantitative data on bulk stoichiometry are actually valuable, and I do not think they may lead to confusion of the reader, quite the opposite, they may help gaining an overall knowledge on the materials. Moreover, knowing bulk stoichiometry may reinforce the importance of surface engineering of the materials. TGA profiles mimicking synthetic conditions, in response to Q4 in my previous report, are valid, and may well be included in the Supplementary Information, with a succinct mention to it, and report of the bulk stoichiometries, in the main text. Other elemental analyses would also be informative. Indium contents for In₂O₃ and In₂O_{2.95} are theoretically 82.7 and 83.0%, respectively. The difference is small, but still discernible by ICP. However, TGA data alone could be sufficient.

8. A brief description of the crystal structure of the stoichiometric In₂O₃ would also prove informative. As reported in the paper suggested by the authors (Chem. Matter. 2014, 26, 5401), and others (e.g. Semiconductor Sci. Tech. 2015, 30, 024001), oxygen vacancies are intrinsic to it.

9. Despite the previous comment, stating that the two observed O 1s XPS signals "corresponded to lattice oxygen and oxygen vacancies" (line 104) is incorrect. They may well correspond to oxygen atoms in octahedral InO₆ environments, and to others next to O vacancies having, as in fact pointed out by the authors in the rebuttal letter to the previous review report (O1s state to decrease by 1.22 eV by loss of a nearby O atom, response to Q6).

10. It is still intriguing why such a small difference in surface O defects content between S3 and S4

results in the observed dramatically enhanced light absorption, and in turn, to 3 orders of magnitude increase in activity. Theoretical calculations suggest the progressive population of mid-gap states, but the reason for the sudden change in properties (and boost in activity) from S3 to S4 requires further explanation.

After carefully going through the author's responses and modifications on the manuscript, I am pleased to check that comparative experiments under monochromatic light have provided evidence for the largely enhanced activity of the material showing clearly superior light absorption. This stronger light absorption is presumably related to the amount of defects, but this does not seem to be a directly proportional relationship, and extended discussion on structure-activity aspects would be appreciated, since it is a central topic of this work. Explanations on O vacancies are convincing in general terms, and the reason why they actually exist in stoichiometric In_2O_3 is clear. However, assignment of spectroscopic signals directly to O vacancies is inaccurate. In addition, brief structural description would help the reader. Despite the noticeable improvement in the soundness of the manuscript, the lattice expansion issue is still unsupported with current data, it cannot be accepted as it is, and moreover, it is not significantly relevant to the main conclusions.

Author Reply: We thank reviewer's positive comments for most of our modifications and deeply appreciate being given the chance to respond to the comments and questions voiced below. All the points were proved to be of great value in producing a much high quality paper, which we hope will be deemed acceptable and enable the paper to be published in Nature Communications

Specific comments:

1. The discussion on lattice expansion. Truly, lattice strain can be used to engineer materials, as in the references suggested by the authors, but always on a short-range. Lattice expansion is only expected for the few layers closer to the amorphous substoichiometric domains, and hence, it is difficult to imagine sufficiently perfect (periodical) expanded lattices, and extending throughout sufficiently large crystalline domains, that would give rise to new, genuine and distinct XRD signals as the shoulders shown in Fig. 1b.

Author Reply: We understand reviewer's concern. As the lattice expansion signal is weak and can only be observed under scale expansion conditions we decided to remove the lattice expansion study from the PXRD section and any mention in the paper has been eliminated. Just for the purpose of scholarship in future studies of $\text{In}_2\text{O}_3\text{-x}$, the Laue Equations, when **the number of unit cells is small, the XRD signal should be visible when the number of unit cells ≥ 5 . The size of the unit cell of In_2O_3 is $a = 0.5487$ nm, $b = 0.5487$ nm, $c = 0.57818$ nm, which means as long as one of the dimensions is larger than 2.9 nm observation of a PXRD reflection is expected.** The spatial extent of our envisioned lattice expanded domain is around 2-4 nm and would be expected to be observable by PXRD.

The Laue Equations describe the intensity of a diffracted peak from a single parallelepiped crystal

$$I = I_e F^2 \frac{\sin^2(\pi/\lambda)(s-s_o) \cdot N_1 a_1}{\sin^2(\pi/\lambda)(s-s_o) \cdot a_1} \frac{\sin^2(\pi/\lambda)(s-s_o) \cdot N_2 a_2}{\sin^2(\pi/\lambda)(s-s_o) \cdot a_2} \frac{\sin^2(\pi/\lambda)(s-s_o) \cdot N_3 a_3}{\sin^2(\pi/\lambda)(s-s_o) \cdot a_3}$$

- N_1 , N_2 , and N_3 are the number of unit cells along the a_1 , a_2 , and a_3 directions
- When N is small, the diffraction peaks become broader
- **The peak area remains constant independent of N**

 Center for Materials Science and Engineering

<http://prism.mit.edu/xray>

Source: <http://prism.mit.edu/XRAY/oldsite/CrystalSizeAnalysis.pdf>

2. Still on lattice expansion, HRTEM measurements carry significant errors. In Fig. 2e, intensity maxima do not exactly fall in the midpoint between two valleys, and if other peak maxima are taken, for example the 6th maximum instead of the 7th, completely different results would be obtained. Higher (even atomic-scale) resolution would be required for lower error and higher reliability of the data.

Author Reply: We agree with reviewer’s point. The ideal situation for studying the lattice expansion process is atomic resolution **in situ EHRTEM**. Unfortunately, to the best of our knowledge, there are only a few institutions that have such equipment and we are not fortunate to be endowed with one. As a result, the lattice expansion study based on our in situ EHRTEM is removed.

3. The lattice spacing data determined for S4 based on the yellow lines in Suppl. Fig. 6 might be biased since such yellow lines do not appear to be orthogonal to the intensity fringes.

Author Reply: After careful reconsideration of the obtained HRTEM images, the image with non-orthogonally aligned arrow is also removed from the study and conclude a lattice expansion in S4 is possible but requires more detailed study by atomic resolution EHRTEM.

4. Lines 177-8: “gradually increasing values 0.292, 0.293, 0.294 and 0.296 nm on passing from S1 through S4” are claimed, but these data are for dynamic measurements on a sample of stoichiometric In₂O₃ under H₂ atmosphere directly on the EHRTEM instrument, not for samples S1 through S4.

Author Reply: This is a fair point - the in situ EHRTEM images represent the measured lattice spacings with increasing treatment time. As mentioned in Q2, we are unable to obtain the atomic resolution level in situ EHRTEM to study the lattice expansion process all mention of which has been removed from the paper.

5. Regarding (2 2 2) spacing values, it is true that a value of around 0.292 nm is consistent with crystallographic data, but it is still intriguing why the spacing remains essentially invariable, or even shows a decreasing trend, when going from S1 through S2 and finally S3, as shown in Suppl. Fig. 5 (0.2918±0.0007 nm, 0.2917±0.0008 nm and 0.2915±0.0005 nm, respectively).

Author Reply: This is a fair point. As mentioned above we do not have access to atomic level EHRTEM measurements so we rounded S1-S3 (222) spacings to 0.292 nm which are the same within experimental error.

6. The theoretical adjacent distance between (2 2 2) facets is reported to be 0.297 nm in Suppl. Fig. 22. This does not match experimental values, as in comment 5 above (0.292 nm).

Author Reply: The simulated crystal structure is based on S4, which has a lattice spacing of 0.296 nm. The simulated lattice spacing of 0.297 nm agrees well with the observed lattice spacing of S4. (only 0.3% error)

7. Quantitative data on bulk stoichiometry are actually valuable, and I do not think they may lead to confusion of the reader, quite the opposite, they may help gaining an overall knowledge on the materials. Moreover, knowing bulk stoichiometry may reinforce the importance of surface engineering of the materials. TGA profiles mimicking synthetic conditions, in response to Q4 in my previous report, are valid, and may well be included in the Supplementary Information, with a succinct mention to it, and report of the bulk stoichiometries, in the main text. Other elemental analyses would also be informative. Indium contents for In₂O₃ and In₂O_{2.95} are theoretically 82.7 and 83.0%, respectively. The difference is small, but still discernible by ICP. However, TGA data alone should be sufficient.

Author Reply: We agree with reviewer’s suggestion and incorporate TGA results for S2 to S4. The following description is added into the manuscript “The thermogravimetric analysis (TGA)

has been performed to simulate the synthetic process of S1 ($2\text{In}(\text{OH})_3 \rightarrow \text{In}_2\text{O}_3 + 3\text{H}_2\text{O}$) and indicates a very similar value of the weight change (84.94%) for S1 to the theoretical stoichiometric In_2O_3 (83.7%), where the very slight excess can be attributed to absorbed water (Supplementary Fig. 3a). Similar measurements were conducted to simulated the synthetic process of S2 to S4 (Supplementary Fig. 3b-d) with the observed weight changes of 99.35%, 99.19%, and 98.63%, and indicates the overall formulas of $\text{In}_2\text{O}_{2.98}$, $\text{In}_2\text{O}_{2.97}$ and $\text{In}_2\text{O}_{2.95}$, respectively. Such minor overall weight change confirmed the formation of [O] and the hydrogenation process is a surface treatment and result in the formation of $\text{In}_2\text{O}_{3-x}@\text{In}_2\text{O}_3$ heterostructures.”

The TGA measurement for S2 to S4 are included in the SI.

8. A brief description of the crystal structure of the stoichiometric In_2O_3 would also prove informative. As reported in the paper suggested by the authors (Chem. Matter. 2014, 26, 5401), and others (e.g. Semiconductor Sci. Tech. 2015, 30, 024001), oxygen vacancies are intrinsic to it.

Author Reply: We agree with reviewer’s suggestion and add the following sentence into the manuscript to strengthen the work. “The cubic bixbyite In_2O_3 is a fluorite-type structure In_2O_3 with $\frac{1}{4}$ of the anions missing indicates a periodic structure that produces 25% structural vacancies.”

9. Despite the previous comment, stating that the two observed O 1s XPS signals “corresponded to lattice oxygen and oxygen vacancies” (line 104) is incorrect. They may well correspond to oxygen atoms in octahedral InO_6 environments, and to others next to O vacancies having, as in fact pointed out by the authors in the rebuttal letter to the previous review report (O1s state to decrease by 1.22 eV by loss of a nearby O atom, response to Q6).

Author Reply: We agree with reviewer’s suggestion and now the XPS peak locates at ~ 531 eV and Raman peak locates ~ 365 cm^{-1} were labeled as the “unsaturated lattice oxygen that generated by the formation of oxygen vacancy (InO_{6-x})”.

10. It is still intriguing why such as small difference in surface O defects content between S3 and S4 results in the observed dramatically enhanced light absorption, and in turn, to 3 orders of magnitude increase in activity. Theoretical calculations suggest the progressive population of mid-gap states, but the reason for the sudden change in properties (and boost in activity) from S3 to S4 requires further explanation.

Author Reply: The local temperature for S1 to S3 can also be estimated via the thermodynamic that is shown in Fig S15. The total yields (ppm) of S1 to S3 are 18, 24 and 50 ppm, which

indicate temperatures even lower than 100 C. The resulted low photothermal local temperatures could only provide limited catalytic performance.

Furthermore, Raman spectroscopy with different beam intensities were performed over S3 and S4 to probe the photothermal effects. As a result, only S4's spectrum exhibits a blue shift with increasing beam intensities and indicates an increasing photothermal local temperatures, and no significant peak shift can be obtained from S3. The related Raman results and explanation were included in the manuscript and SI.

The added explanation includes “The reason for such an impressive photo-enhancement can be attributed to the much stronger solar energy harvesting ability and photothermal effects of S4 compared to the other samples (Supplementary Fig. 7a and Supplementary Fig.8), which results in a larger photothermal effect and correspondingly higher catalytic performance. The local temperatures of all samples can be estimated from the conversion of CO₂ to CO (yield, ppm), whereas S4 can be calculated as 262 °C and S1 to S3 have much lower local temperatures, Supplementary Fig. 15.”, and “Further Raman spectroscopy with different beam intensities were conducted over S3 and S4 to study their photothermal effects. The resulting Raman signals for S4 exhibit blue shift with increasing beam intensities (2.17, 5.44, 10.85 and 21.66 mW/μm) gradually shift from 304.18 to 302.90, 302.68 and 300.63 cm⁻¹ and indicates an increasing trend of photothermal local temperatures. On the contrary, no significant shift can be observed from S3 and implies minor photothermal effects.”.

REVIEWERS' COMMENTS:

Reviewer #1 (Remarks to the Author):

The last round of modifications has addressed the most serious concerns related to it. Reasons for improved activity are now clearer and sounder. Perhaps, some further explanations and evidence on the photothermal effects could be included. Aspects to consider:

1. More solid evidence supporting the claimed contribution of photothermal effects on the impressively increased activity of S4 are required. For example, direct local temperature measurements on the surface of the catalysts under operation by using contact (thermocouple) measurements, or the use of thermal cameras would be truly informative. The method based on the expected equilibrium composition is an approximation, but ignores the possibility of any photonic mechanism, or of kinetic or mass transfer constraints.
2. Regarding the use of Raman to prove the photothermal effect, the related discussion should be extended. The assignment of the signal selected to track the effect of increasing irradiance must be suggested/commented, and the observed frequency shift made more clearly visible by expanding the corresponding areas for both S3 and S4 (Figures 8c and d).
3. Irradiance in Raman experiments appear to be extremely high: $2\text{-}20 \text{ mW}/\mu\text{m}^2 = 2\text{-}20 \cdot 10^9 \text{ W}/\text{m}^2$, that is, $2\text{-}20 \cdot 10^6$ suns, more than one million times solar irradiance. Is that correct? Moreover, the units in line 136 are wrong: " μm^2 ", NOT " μm ".
4. It is clear that substoichiometric domains are formed on the surface of the parent In_2O_3 , as expected, and suggested by different techniques. Then, the notation " $\text{In}_2\text{O}_3\text{-x@In}_2\text{O}_3$ " does not appear correct if one visualises "@" as indicating encapsulation or inclusion. In my opinion, " $\text{In}_2\text{O}_3\text{-x/In}_2\text{O}_3$ " is more appropriate.

REVIEWERS' COMMENTS:

Reviewer #1 (Remarks to the Author):

The last round of modifications has addressed the most serious concerns related to it. Reasons for improved activity are now clearer and sounder. Perhaps, some further explanations and evidence on the photothermal effects could be included. Aspects to consider:

Author Reply: We deeply appreciate the reviewer's positive comments for our modifications and grateful for being given the chance to respond to the last few comments and questions voiced below. All the points raised proved to be of great value in producing a much high quality paper, which we hope will be deemed acceptable and enable the paper to be published in Nature Communications without too much further delay.

1. More solid evidence supporting the claimed contribution of photothermal effects on the impressively increased activity of S4 are required. For example, direct local temperature measurements on the surface of the catalysts under operation by using contact (thermocouple) measurements, or the use of thermal cameras would be truly informative. The method based on the expected equilibrium composition is an approximation, but ignores the possibility of any photonic mechanism, or of kinetic or mass transfer constraints.

Author Reply: We agree with reviewer's suggestion and added the thermocouple readings and additional explanation into the manuscript.

"Based on the enclosed thermocouple, the temperatures of S1 to S3 are lower than 50 °C, and about 160 °C for S4. The local temperatures of all samples can be estimated from the conversion of CO₂ to CO (yield, ppm), where S4 is found to be 262 °C in contrast to S1 to S3 which are found to have much lower local temperatures, **Supplementary Fig. 15**. These results illustrate photocatalysis and photothermal catalysis can be achieved with light irradiation and in this case serve to shift the reaction equilibrium equivalent to one corresponding to 262 °C."

2. Regarding the use of Raman to prove the photothermal effect, the related discussion should be extended. The assignment of the signal selected to track the effect of increasing irradiance must be suggested/commented, and the observed frequency shift made more clearly visible by expanding the corresponding areas for both S3 and S4 (Figures 8c and d).

Author Reply: We agree with reviewer's suggestion and added additional explanation.

"The Raman signal located at ~308 cm⁻¹ has been assigned to a vibrational/phonon mode of an InO₆ site and is used as the probe/reference signal."

3. Irradiance in Raman experiments appear to be extremely high: 2-20 mW/μm² = 2-20 10⁹

W/m², that is, 2-20 10⁶ suns, more than one million times solar irradiance. Is that correct? Moreover, the units in line 136 are wrong: “μm²”, NOT “μm”.

Author Reply: We are thankful for the reviewer’s question and realize now an error we made in conversion units. The beam intensities are now corrected as 0.00110, 0.00055, 0.00028, and 0.00011 mW/μm².

4. It is clear that substoichiometric domains are formed on the surface of the parent In₂O₃, as expected, and suggested by different techniques. Then, the notation “In₂O₃-x@In₂O₃” does not appear correct if one visualises “@” as indicating encapsulation or inclusion. In my opinion, “In₂O₃-x/In₂O₃” is more appropriate.

Author Reply: We agree with the reviewer’s suggestion and the notation for black indium oxide has now been corrected as “In₂O₃-x/In₂O₃”.